EMBO
Molecular Medicine

# Improved gene therapy for spinal muscular atrophy in mice using codon-optimized hSMN1 transgene and hSMN1 gene-derived promotor

Qing Xie [1,2,7], Xiupeng Chen [1,2,7], Hong Ma[1,3], Yunxiang Zhu[4], Yijie Ma [4], Leila Jalinous[4], Gerald F Cox[4], Fiona Weaver[4], Jun Yang[4], Zachary Kennedy [4], Alisha Gruntman[1,5], Ailing Du[1,2], Qin Su[1,2,3], Ran He[1,2,3], Phillip WL Tai [1,2,6], Guangping Gao [1,2,6✉] & Jun Xie [1,2,3✉]

## Abstract

**Physiological regulation of transgene expression is a major challenge in gene therapy. Onasemnogene abeparvovec (Zolgensma®) is an approved adeno-associated virus (AAV) vector gene therapy for infants with spinal muscular atrophy (SMA), however, adverse events have been observed in both animals and patients following treatment. The construct contains a native human survival motor neuron 1 (*hSMN1*) transgene driven by a strong, cytomegalovirus enhancer/chicken β-actin (*CMVen/CB*) promoter providing high, ubiquitous tissue expression of SMN. We developed a second-generation AAV9 gene therapy expressing a codon-optimized *hSMN1* transgene driven by a promoter derived from the native *hSMN1* gene. This vector restored SMN expression close to physiological levels in the central nervous system and major systemic organs of a severe SMA mouse model. In a head-to-head comparison between the second-generation vector and a benchmark vector, identical in design to onasemnogene abeparvovec, the 2nd-generation vector showed better safety and improved efficacy in SMA mouse model.**

**Keywords** AAV; SMA; SMN1; Gene Therapy; Endogenous Promoter
**Subject Categories** Genetics, Gene Therapy & Genetic Disease; Musculoskeletal System

See also: MM Zwartkruis & EJN Groen

## Introduction

Spinal muscular atrophy (SMA) is an autosomal recessive disease characterized by progressive degeneration of α-motor neurons located in the anterior horns of the spinal cord, which results in symmetrical muscle weakness and atrophy. The disease is caused by loss of function mutations in the survival motor neuron 1 gene (*SMN1*) encoding SMN. SMN plays important roles in pre-mRNA splicing, assembly of small nuclear ribonucleoprotein particles, mRNA transport in neuronal axons, and protein translation (Bernabo et al, 2017; Hamilton and Gillingwater, 2013). *SMN1* has a paralog, *SMN2*, that is unable to compensate for the loss of *SMN1* because the majority of *SMN2* transcripts (~85%) lack exon 7 due to a spice site mutation, resulting in non-functional SMN protein (Lefebvre et al, 1995; Lorson et al, 1999). Although *SMN2* expresses only a small amount of functional SMN protein (~15%), the number of *SMN2* gene copies varies in the population and is a major phenotypic modifier of disease severity, with more copies being associated with less severe phenotypes. SMA affects approximately 1 in 11,000 newborns and remains the leading genetic cause of infant deaths (Crawford and Pardo, 1996; Lefebvre et al, 1995; Lunn and Wang, 2008).

There are currently three widely approved therapies for SMA that all act to increase SMN protein levels either by enhancing the inclusion of exon 7 during endogenous *SMN2* pre-mRNA splicing or by introducing an exogenous *SMN1* gene (Jablonka et al, 2022; Singh et al, 2017; Sumner and Crawford, 2018). The intrathecally delivered antisense oligonucleotide, nusinersen (Spinraza®) (Darras et al, 2019; Finkel et al, 2017; Hua et al, 2007; Hua et al, 2008; Ramos et al, 2019; Singh et al, 2009) and the oral small molecule, risdiplam (Evrysdi®) (Darras et al, 2021; Gowda et al, 2022; Singh et al, 2020), are splicing modulators that increase the amount of SMN2-derived transcripts that retain exon 7 and are approved for pediatric and adult SMA patients. Onasemnogene abeparvovec (Zolgensma®), a self-complementary adeno-associated virus serotype 9 (scAAV9) vector that expresses the native *SMN1* transgene,

[1]Horae Gene Therapy Center, UMass Chan Medical School, Worcester, MA, USA. [2]Department of Microbiology and Physiological Systems, UMass Chan Medical School, Worcester, MA, USA. [3]Viral Vector Core, UMass Chan Medical School, Worcester, MA, USA. [4]CANbridge Pharmaceuticals, Burlington, MA, USA. [5]Pediatrics, UMass Chan Medical School, Worcester, MA, USA. [6]Li Weibo Institute for Rare Diseases Research, UMass Chan Medical School, Worcester, MA, USA. [7]These authors contributed equally: Qing Xie, Xiupeng Chen. ✉E-mail: guangping.gao@umassmed.edu; jun.xie@umassmed.edu

is an intravenously administered gene therapy that is approved for SMA patients with biallelic mutations in the *SMN1* gene who are less than 2 years of age (Mendell et al, 2017; Strauss et al, 2022). Although no head-to-head clinical trials have been performed, indirect comparisons suggest that onasemnogene abeparvovec may outperform nusinersen in event-free survival, overall survival, dependence on permanent assisted ventilation, and motor milestone achievements in type I SMA patients (Bischof et al, 2021; Dabbous et al, 2019).

Adeno-associated virus (AAV) belongs to the *Parvoviridae* family of DNA viruses and has been widely used in clinical trials and clinical applications as an effective gene transfer vehicle (Mendell et al, 2021). As a gene therapy, intravenously administered AAV9 efficiently transduces peripheral organs and crosses the blood-brain barrier to transduce cells of the central nervous system (CNS) (Foust et al, 2009; Pattali et al, 2019; Zhang et al, 2011). Currently, over 3000 patients have been treated with onasemnogene abeparvovec (NOVARTIS 2022a). The high dose required for efficacy, $1.1 \times 10^{14}$ vector genomes/kg of body weight, however, has been associated with safety concerns, most notably a black box warning in the US package insert for severe liver injury and acute liver failure (Mendell et al, 2017; Strauss et al, 2022). Increase liver transaminases were reported as an adverse event in 27.3% of 44 clinical trial subjects reported in the package insert (Mendell et al, 2017; Strauss et al, 2022). In a meta-analysis of clinical safety data conducted by Novartis on onasemnogene abeparvovec through 2019, liver-associated adverse events were reported in 34% of 100 patients across five clinical trials and in 23% of 43 patients in a managed access program and diseases registry (Chand et al, 2021). To date, two patients have died from acute liver failure (NOVARTIS 2022b). A fatal case of systemic thrombotic microangiopathy was reported after a child, carrying a predisposing risk factor in the *complement factor I* gene, received onasemnogene abeparvovec (Guillou et al, 2022). Most recently, a 3-year-old child developed hemophagocytic lymphohistiocytosis after treatment with onasemnogene abeparvovec (Galletta et al, 2022).

One possible mechanism underlying these adverse events is the immune response to the AAV9 capsid as a result of the high vector dose required for efficacy. Another possibility is the ubiquitous and constitutive expression of *SMN1* driven by the strong cytomegalovirus enhancer/chicken β-actin (*CMVen/CB*) promoter. SMN protein is normally expressed throughout the body at varying levels in organs and tissues, and severe alterations in expression levels contributes to pathologies not only within the central nervous system, more specifically within motor neurons, but also in other organs, including liver, heart, pancreas, bone, and intestine (Besse et al, 2020; Chand et al, 2021; Gombash et al, 2015; Heier et al, 2010; Hua et al, 2011; Lunn and Wang, 2008; Nash et al, 2016; Rudnik-Schoneborn et al, 2008; Schreml et al, 2013; Shababi et al, 2010). In pre-clinical studies, administration of high doses of AAV gene therapies, including SMN, were found to cause liver toxicities and neurodegeneration of the dorsal root ganglia (DRG) in nonhuman primates (NHPs) (Hinderer et al, 2018; Hudry et al, 2023; Tukov et al, 2022). Recently, a long-term study in SMA mice (SMNΔ7 mice) suggested that rAAV9-mediated overexpression of hSMN by another ubiquitous promoter (GUSB) leads to the loss of proprioceptive neurons in DRG and motor neurons, although the exact mechanism of the observed toxicities is unknown (Van Alstyne et al, 2021).

We hypothesized that the use of a lower vector dose with a codon-optimized *SMN1* transgene (*co-hSMN1*) would restore SMN expression close to physiological levels in relevant organs, providing an improved risk to benefit ratio for SMA patients. Initially, we designed a vector with the *co-hSMN1* transgene driven by the *CMV enhancer/CB* promoter packaged in AAV9 (rAAV9-*CMVen/CB-co-hSMN1*). However, treatment of SMNΔ7 mice with this vector resulted in early mortality as a consequence of supraphysiological levels of *hSMN1* in the liver. To achieve physiologically regulated *hSMN1* expression throughout the body, we created a second-generation scAAV9 vector that expressed *co-hSMN1* under the control of a promoter derived from the endogenous *hSMN1* promoter (scAAV9-*SMN1p-co-hSMN1*). A head-to-head comparison between this 2nd-generation vector and a benchmark vector (BMK), whose vector genome is identical in design to onasemnogene abeparvovec (scAAV9-*CMVen/CB-hSMN1*), revealed therapeutic advantages of the 2nd-generation vector, both in terms of better safety (no increase in transaminase levels) and improved efficacy (e.g., lower effective dose; broader therapeutic window; longer life span; increased cardiac, respiratory, and motor functions; and minimal peripheral disease manifestations).

## Results

### Supraphysiological SMN expression from initial vector design contributed to early mortality and liver abnormalities

We first generated a benchmark construct containing the human *SMN1* transgene driven by the *CMVen/CB-hSMN1* promoter. This construct has an identical vector genome design to onasemnogene abeparvovec (Kaspar and Foust 2009), differing only by a cloning site next to the mutant ITR (Fig. 1A; Appendix Fig. S1). To decrease adverse events by administering a lower dose of vector while maintaining or improving therapeutic outcomes, we replaced the original native *SMN1* coding sequence (*hSMN1*) with a codon-optimized human *SMN1* (*co-hSMN1*) sequence. In addition, a synthetic intron was placed in front of the transgene to enhance expression (Fig. 1A). This vector, hereafter referred to as vector 1, and the benchmark vector were used to transfect cultured mouse Neuro2a cells, which revealed these optimization steps resulted in a 3-fold increase in SMN protein level compared to the benchmark vector (Fig. 1B). The constructs were then packaged into AAV9 and injected intravenously into SMNΔ7 mice, which is a severe SMA mouse model that harbors a single targeted null mutation (*SMN*$^{-/-}$) and two transgenic alleles (*hSMN2*$^{+/+}$ *and SMNΔ7*$^{+/+}$), recapitulating many of the clinical features observed in SMA type 1, such as motor neuron loss, motor function deficits, and early death. Facial vein injection of the benchmark vector into postnatal Day 0 (P0) mice with a single dose of 3.3E + 14 vg/kg increased the median lifespan of SMA mice from 14 days to 58 days. The initial *co-hSMN1* vector (Vector 1) at the same dose failed to extend the lifespan of SMA mice, and ultimately led to early death. After lowering the dose of Vector 1 to 1.1E + 14 vg/kg and 0.5E + 14 vg/kg, the median lifespan increased to 16 and 27 days, respectively (Fig. 1C). Treatment of mice with another codon-optimized vector (Vector 2), which contained the same *CMVen/CB* promoter and *co-hSMN1* transgene but without the synthetic intron, also showed increased median lifespans at lower doses, but not at the high dose

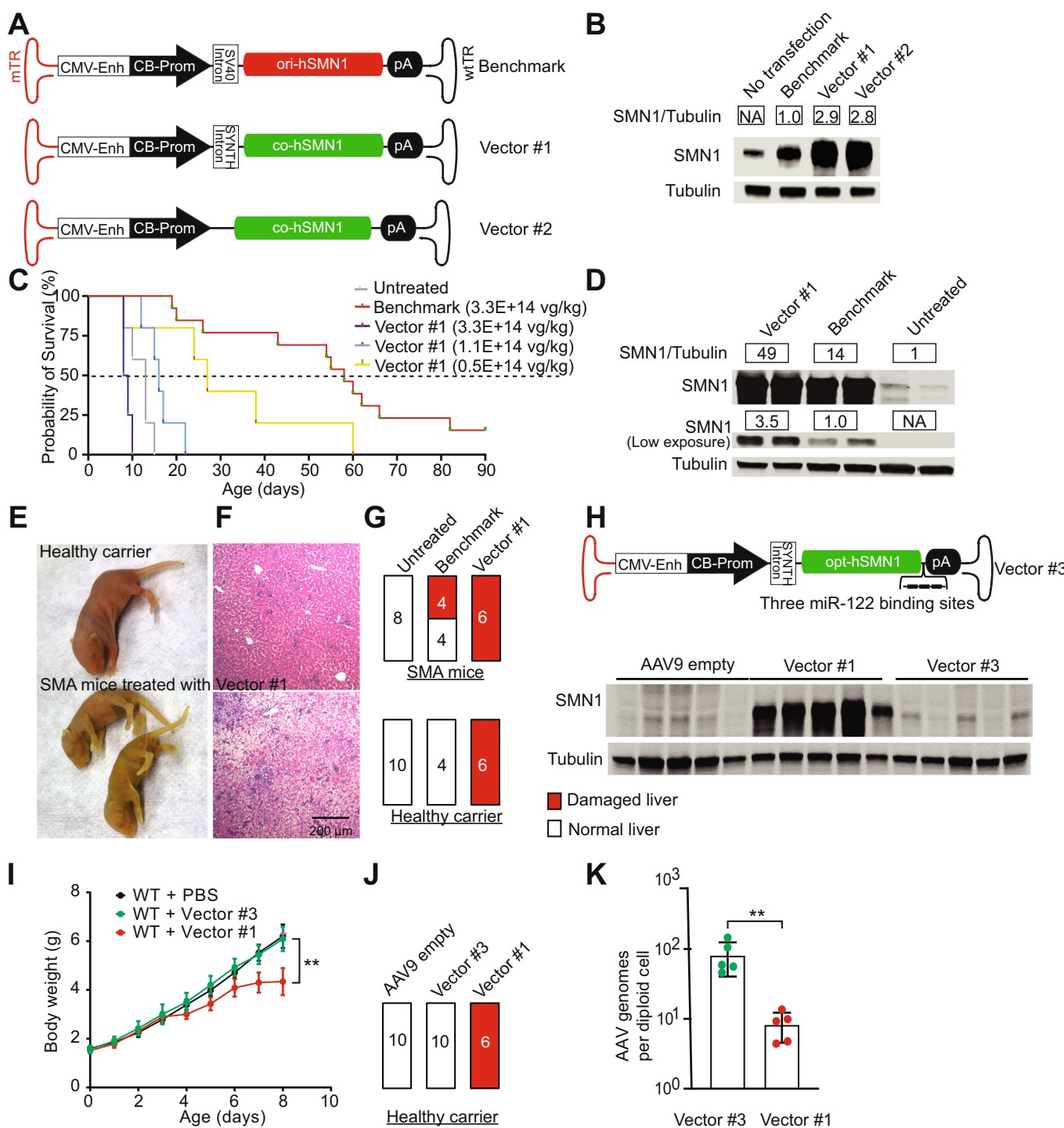

© This is a U.S. Government work and not under copyright protection in the US; foreign copyright protection may apply    EMBO Molecular Medicine    Volume 16 | April 2024 | 945–965    **947**

of 3.3E + 14 vg/kg (Fig. EV1). Overall, no survival benefit was observed with the initial *co-hSMN1* vectors (Vectors 1 and 2) compared to the benchmark vector.

In all SMA mice that received the high dose (3.3E + 14 vg/kg) of Vector 1, we noted jaundice, an indicator of hepatotoxicity, that lasted for approximately one week (Fig. 1E). We sacrificed the animals at P8 for further evaluation. Western blot analysis showed that the SMN protein level in the liver of the mice receiving Vector

1 was 3.5-fold higher than in the benchmark vector-treated animals. Compared to the untreated animals, the SMN expression in liver was increased 49- and 14-fold by Vector 1 and benchmark vector, respectively (Fig. 1D). Hematoxylin and eosin (H&E) staining revealed severe necrosis in the livers of all animals that received the high dose of Vector 1, including both SMA mice and their healthy littermates (Fig. 1F,G). We hypothesized that supraphysiological levels of SMN in liver, produced by Vector 1,

**Figure 1.  Supraphysiological levels of SMN contributes to hepatotoxicity in mice.**

(A) Schematic representation of the designed plasmids. A vector genome identical to onasemnogene abeparvovec was produced as the benchmark vector. A codon-optimized human *SMN1* gene (*co-hSMN1*) was synthesized and incorporated into a scAAV genome with the cytomegalovirus enhancer/chicken β-actin promoter (*CMVen/CB-hSMN1*) in the presence or absence of a synthetic intron as Vector 1 and Vector 2, respectively. (B) A representative image of Western blot ($n = 3$) of Neuro2a cells transfected with the plasmids in (A). (C) Kaplan-Meier survival curves of homozygous SMNΔ7 mice injected at P0 via the facial vein at three different doses ($3.3E + 14$ vg/kg, $n = 12$; $1.1E + 14$ vg/kg, $n = 7$; $0.5E + 14$ vg/kg, $n = 5$). The benchmark vector was injected as a reference at a dose of $3.3E + 14$ vg/kg ($n = 13$). Non-injected SMNΔ7 mice ($n = 4$) were used as controls. (D) Western blot analysis of SMN expression in mouse liver 8 days after injection of Vector 1 and the benchmark vector. (E) Photographs of neonatal mice 8 days after injection. (F) H&E staining was performed to show liver pathology. Bar = 200 μm. (G) Both SMA mice and healthy carriers were injected at P0 with $3.3E + 14$ vg/kg of Vector 1 or the benchmark vector. Liver histopathology was analyzed by H&E staining for each animal 8 days post-injection. Animals with damaged livers were counted. (H) The rAAV-cohSMN1 vector construct with reduced SMN expression in liver contains three copies of a liver-specific, miR-122 binding site in the Vector 1 genome and is packaged in AAV9 (top). SMA mice were treated at P0 by facial vein injection at a dose of $3.3E + 14$ vg/kg, and at eight days post-injection, mouse livers were harvested for Western blot analysis (bottom). (I) Animal weights were measured daily during the 8-day study period ($n = 8$ in the PBS group; $n = 10$ in Vector #3 group and $n = 7$ in Vector #1 group). Error bars, s.d. (J) Pathological evaluation of liver, by H&E staining was conducted to determine the number of animals with liver damage, and (K) vector genome copy number ($n = 5$ in each group). Student's t-test. **$P < 0.01$. Error bars, s.d. Source data are available online for this figure.

were responsible for the severe hepatotoxicity. To test this hypothesis, a new construct was created (Vector 3) by incorporating three miR-122 binding sites into the 3'-UTR of the *co-hSMN1* vector to minimize hepatocyte expression (Fig. 1H). In the presence of liver-specific miR-122, the 3'-UTR of SMN1 mRNA is cleaved at the miR-122 binding sites, thereby suppressing transgene expression in a liver-specific manner (Xie et al, 2011). Vector 3 was injected into P0 neonatal mice at a dose of $3.3E + 14$ vg/kg, and we observed a substantial reduction in SMN expression within the liver (Fig. 1H). Both body weight loss and hepatotoxicity were alleviated by suppressing SMN expression in the liver (Fig. 1I,J). Although both vectors (Vector 1 and Vector 3) were injected at the same dose ($3.3E + 14$ vg/kg), substantially fewer AAV genomes were detected on Day 8 post-injection in the livers of mice that received Vector 1, which lacks the miR-122 binding sites, compared to Vector 3-treated mice ($8.4 \pm 3.8$ versus $82.6 \pm 42.1$ vg/diploid cell, respectively). This finding suggested that the observed hepatotoxicity was followed by liver regeneration which ultimately led to vector genome dilution (Fig. 1K). Collectively, these results indicated that the toxicity associated with supraphysiological expression of SMN can compromise the therapeutic outcome of AAV-mediated gene therapy and a solution would be to engineer a vector that produces more physiological levels of SMN across organs and tissues.

## Endogenous *SMN1* promoter-driven *SMN1* gene replacement therapy substantially prolonged the lifespan of SMA mice beyond that of benchmark vector-treated mice

Overexpression or deficiency of SMN in many tissues may lead to unknown toxicities or disease complications. Due to the wide tissue tropism of AAV9 and the constitutive potency of the *CMVen/CB* promoter incorporated into the benchmark vector, safety concerns may arise as a result of SMN overexpression, which could impact various organs/tissues, such as heart, skeletal muscle, and DRG. Whole-body restoration of SMN expression to physiological levels would be desirable for a safe and effective gene therapy.

Using the *co-hSMN1* vector (Vector 1), lacking the miR-122 binding sites, we replaced the ubiquitous *CMVen/CB* promoter and synthetic intron with a derivative of the endogenous promoter of the *hSMN1* gene (Echaniz-Laguna et al, 1999). The SMN1 endogenous promoter ($-767$ to $+152$) was selected because of its efficient promoter activity in the cultured cells and proper size to fit into a

scAAV genome which has ~2.5 kb packaging limit. We note that the SMN promoter sequence that was previously described has since been revised by the sequencing of the full human genome. Whole epigenome analyses and enhancer/promoter prediction has also been performed at all annotated human genes. Interestingly, enrichment of epigenetic marks (H3K27me3, H3K36me3, H3K4me1, and H3K9me3) are all within the $-767$ to $+152$ region as demonstrated in neuroblast cells (BE2C) (Appendix Fig. S2A). This finding suggests that the SMN promoter used in our cassette encompasses all of the regulatory elements predicted to be near the transcriptional start site. Furthermore, there are only ten differences in the SMN promoter previously reported and the most recent build (T2T CHM13v2.0/hs1) (Appendix Fig. S2B). Nevertheless, this new construct displayed a comparable expression to Vector 1 in Neuro2a cells (Fig. 2A). We packaged this new construct in an AAV9 capsid (scAAV9-SMNp-cohSMN1), henceforth referred to as the 2nd-generation vector, and injected it into P0 SMA mice at three different doses ($3.3E + 14$ vg/kg, high dose; $1.1E + 14$ vg/kg, medium dose; and $0.5E + 14$ vg/kg, low dose) via facial vein administration and observed the mice for a period of 90 days (Fig. 2B). In SMA mice treated with the 2nd-generation vector, survival to Day 90 was 100% (10/10) at $3.3E + 14$ vg/kg, 93% (13/14) at $1.1E + 14$ vg/kg, and 38% (3/8) at $0.5E + 14$ vg/kg. In contrast, only 13% (2/15) of mice that received the benchmark vector at $3.3E + 14$ vg/kg survived to Day 90, and no mice (0/10) that received $1.1E + 14$ vg/kg survived (Fig. 2C). The 2nd-generation vector significantly extended the life span of SMA mice in a dose-dependent manner, with all doses showing improved survival compared to the benchmark vector or to no treatment (p < 0.01 for all comparisons). Importantly, the middle dose ($1.1E + 14$ vg/kg) of the 2nd-generation vector showed substantially better survival than the high dose ($3.3E + 14$ vg/kg) of the benchmark vector with a median survival of 90 days vs 55 days, respectively (Fig. 2C). The body weight gain conferred by the high dose of the 2nd-generation vector, which reached ~80% of healthy carrier littermates, was greater than the weight gain achieved by the high dose of the benchmark vector (Fig. 2D). The middle dose of the 2nd-generation vector led to body weight gain that was comparable to that of the high dose of the benchmark vector (Fig. 2D).

Since early treatment has been shown to improve SMA type 1 outcomes (Sumner and Crawford, 2022), sought to determine whether administration of the 2nd-generation vector at a later timepoint would maintain therapeutic efficacy. Therefore, we injected the benchmark and 2nd-generation vectors into postnatal

 

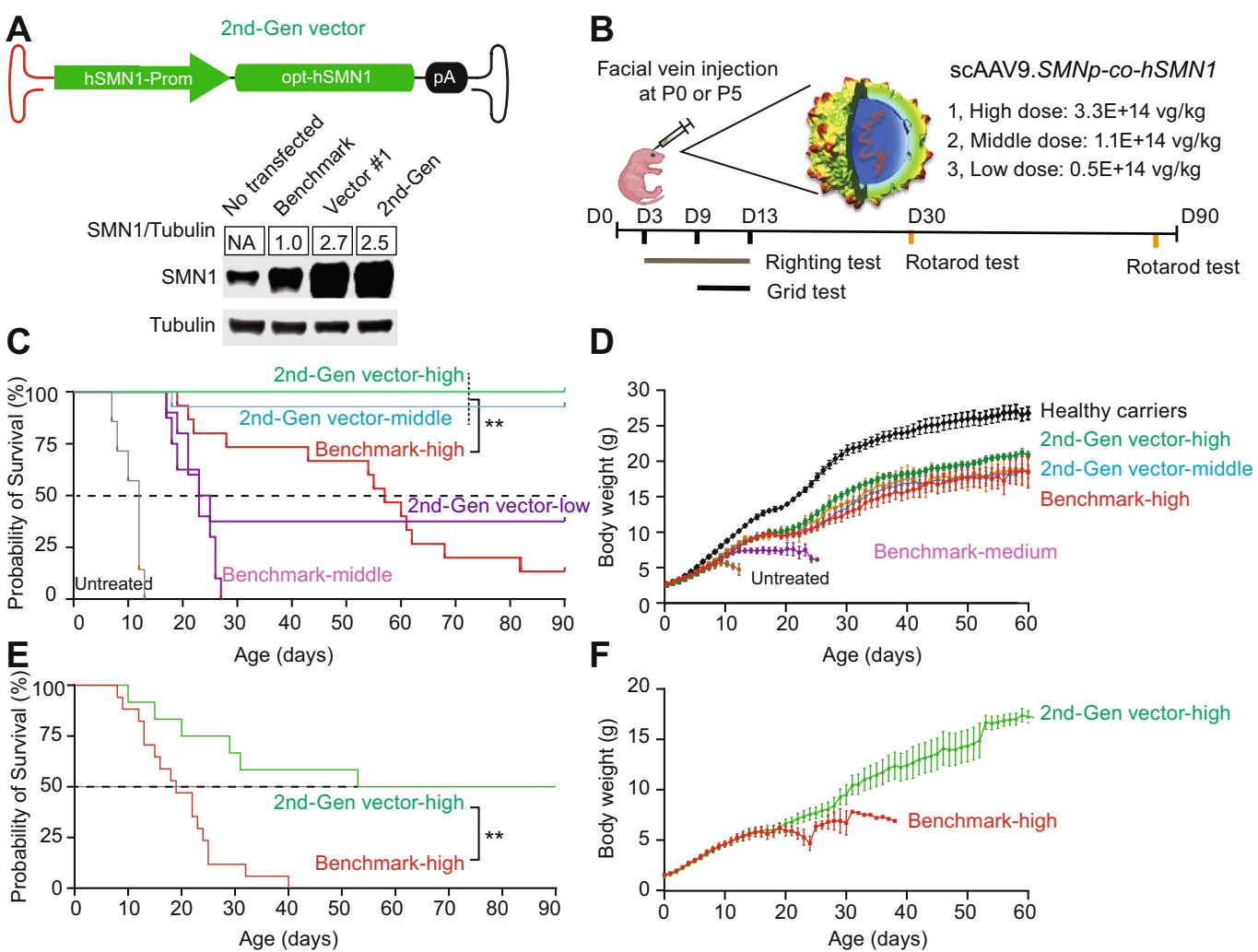

**Figure 2. The 2nd-generation vector greatly improved the lifespan and body weight gain of SMA mice.**

(A) The 2nd-generation vector containing the endogenous *hSMN1* promoter and codon-optimized human *SMN1* coding sequence (*co-hSMN1*) (top). Western blot analysis of SMN expression in Neuro2a cells transfected with the benchmark vector, Vector 1, and the 2nd-generation vector plasmid (bottom), $n = 3$. (B) Schematic diagram of the animal study. SMA mice were injected via the facial vein with the 2nd-generation vector at three different doses (high, $3.3 + 14$ vg/kg, $n = 10$; middle, $1.1 + 14$ vg/kg, $n = 13$; low, $0.5 + 14$ vg/kg, $n = 8$) or benchmark vector at $3.3 + 14$ vg/kg ($n = 15$) or $1.1 + 14$ vg/kg ($n = 10$) at P0. The surface righting tests were performed on Days 3–13, the grid tests were conducted on Days 9–11, and the rotarod tests were conducted on Days 30 and 90 for all surviving mice. (C, D) Survival and body weight gain of SMA mice injected at P0. Non-injected healthy carriers were used as controls ($n = 12$). (E, F) Survival and body weight gain of SMA mice injected at P5 ($n = 17$ in the benchmark group and $n = 12$ in the 2nd-generation vector group). Body weight data represent the mean ± S.E.M. Survival curve statistical analysis was performed using a Mantel-Cox test. \*\*$P < 0.01$ Source data are available online for this figure.

Day 5 (P5) SMA mice at the same dose of $3.3 + 14$ vg/kg. The median survival of SMA mice treated with the benchmark vector was 19 days versus 69 days for the 2nd-generation vector ($p < 0.01$) (Fig. 2E), with half of the latter animals ($n = 12$) surviving until the end of the study (Day 90). Although body weight gain in mice treated at P5 was slower at the beginning (within 30 days post-injection) compared to P0-treated mice, body weight steadily increased and reached ~18 grams by two months of age which was comparable to the weight of P0-treated mice (~20 g) (Fig. 2F). Together, these results indicate that the 2nd-generation vector, using the endogenous *SMN1* promoter to regulate *SMN1* gene expression increased survival, and had a broader therapeutic time window compared to the benchmark vector.

## The 2nd-generation vector restored motor function in SMA mice more efficiently than the benchmark vector

We next aimed to assess motor function in SMA mice treated at P0 with the 2nd-generation vector at a dose of $3.3 + 14$ vg/kg. All healthy carriers were able to right by day 3 and that untreated SMA mice began to right at day 7. SMA mice treated with the 2nd-generation vector started to right themselves as soon as Day 3 after injection, and all could right themselves by Day 7, whereas the benchmark vector-treated animals started to right themselves on Day 7, and all could right themselves by Day 13 (Fig. 3A). The 2nd-generation vector-treated animals performed as well as healthy carriers on the grid test at Day 11 (49.3 s vs 59.1 s, respectively) and

 

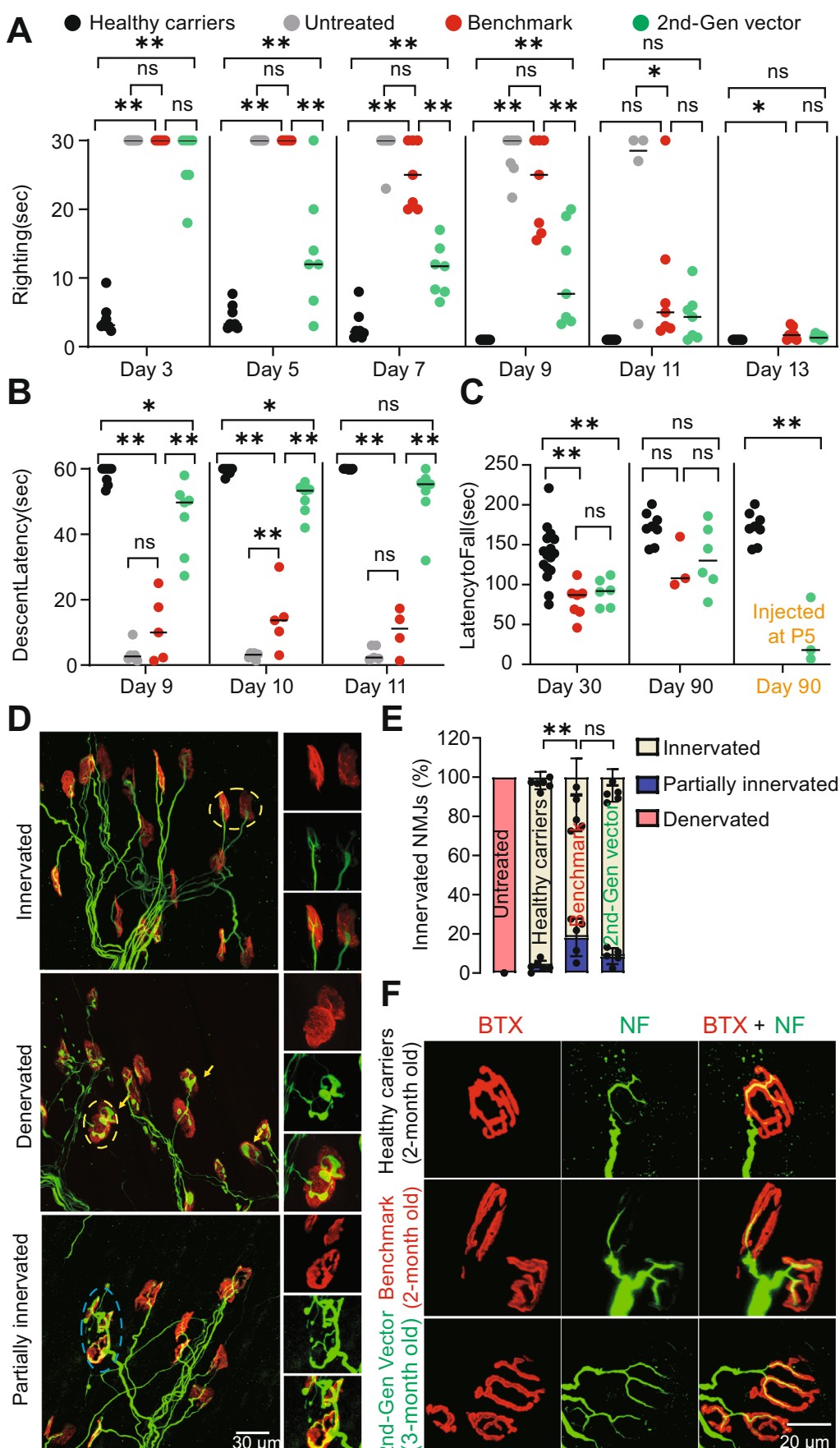

**Figure 3.   The 2nd-generation vector improved motor function in SMA mice.**

(A) Surface righting reflex of treated SMA animals ($n = 7$ in the benchmark group and $n = 7$ in 2nd-Gen vector group) with healthy littermates ($n = 8$) and non-injected SMA mice ($n = 10$) as controls. (B) Grid tests of treated SMA animals ($n = 5$ in the benchmark group and $n = 7$ in 2nd-Gen vector group) with healthy littermates ($n = 8$) and non-injected SMA mice ($n = 6$) as controls. (C) Rotarod testing of SMA mice treated at P0 ($n = 16$ in the healthy littermate group; $n = 6$ in both the benchmark and 2nd-Gen vector group on day 30. On day 90, $n = 8$ in the healthy littermate group; $n = 3$ in the benchmark group and $n = 6$ in the 2nd-Gen vector group) and P5 ($n = 8$ in the healthy littermate group and $n = 3$ in the benchmark group). Bars denote the median. (D) NMJ evaluation of mice at 12 days of age. TVA muscles were isolated from Day 12 postnatal mice for immunostaining. Neurofilament (NF) was stained using the Alex 488 fluorescence antibody (green), and neuromuscular junctions were visualized by the Alexa Fluor™ 594 conjugated α-bungarotoxin (BTX) reagent (red). The yellow arrows indicate accumulation of neurofilament, the yellow dashed circle indicates a collapsed NMJ structure, and the blue dashed circle indicates a partially innervated NMJ structure. (E) Quantification of the frequency of innervated, partially innervated, and denervated NMJs among the indicated groups. Five animals were included for each vector-treated group, and three animals were included for untreated SMA mice. Data represents the mean ± SD. (F) NMJ evaluation of mice at 2-3 months of age. One-way ANOVA. *$P < 0.05$; **$P < 0.01$; ns, not significant. Source data are available online for this figure.

significantly outperformed the benchmark vector group at all three timepoints (Days 9, 10, and 11) ($p < 0.01$ for all comparisons) (Fig. 3B). Mice injected with the 2nd-generation or benchmark vector at P0 performed comparably during rotarod testing on Day 30 ($78.5 \pm 21.2$ s vs $90.3 \pm 17.2$ s, respectively) and Day 90 ($122.7 \pm 32.6$ s vs $133.3 \pm 40.7$ s, respectively). However, both performed worse than healthy carriers on both Days 30 and 90 ($137.8 \pm 34.5$ s and $171.6 \pm 19.6$ s, respectively). Furthermore, SMA mice treated with the 2nd-generation vector at P5 performed significantly worse compared ($36.3 \pm 41.6$ s, $p < 0.01$) to P0 2nd-generation vector-treated mice, indicating the importance of early intervention (Fig. 3C). In summary, animals treated with the 2nd-generation vector demonstrated rapid and marked improvement in overall muscle strength and coordination compared to the benchmark vector-treated mice, performing at comparable levels to healthy carriers when treated at P0.

We next examined the neuromuscular junctions (NMJs) of the transversus abdominis (TVA) muscle, which are known to be severely affected in SMA mice by P12 (Murray et al, 2008). Immunostaining of TVA muscle in healthy carriers showed no aberrant NMJ structures (Fig. 3D, top) while untreated SMA mice revealed 100% denervation of NMJs (Fig. 3E). Furthermore, typical neurofilament accumulation, collapsed NMJ structures (Fig. 3D, middle), and partially innervated NMJs were observed in the treated mice (Fig. 3D, bottom). Only $8.6 \pm 4.1\%$ of NMJs in the 2nd-generation vector treatment group were partially innervated, which was similar to that observed in healthy carriers ($3.5 \pm 2.7\%$, $p = 0.47$), whereas benchmark-treated mice had significantly more partially innervated NMJs compared to healthy carriers ($18 \pm 9.5\%$, $p < 0.01$) (Fig. 3E). By the end of the study (Day 90), the 2nd-generation vector-treated mice had better-appearing NMJ architecture compared to the benchmark vector-treated (Fig. 3F). Improvement in NMJ structure conferred by the 2nd-generation vector was consistent with the improvement in motor function observed in SMA mice.

## The 2nd-generation vector resulted in physiological levels of transaminases, and rescued heart and respiratory functions in SMA mice better than the benchmark vector

Whole-body plethysmography was used to evaluate respiratory function. One month after injecting SMA mice with a dose of $3.3\mathrm{E} + 14$ vg/kg, we observed rapid breathing in the benchmark vector group, but not in the 2nd-generation vector group (Video EV 1). In the 2nd-generation vector-treated mice, we also observed improved respiratory function with the trends toward increased tidal volume and peak inspiratory flow and a lower respiratory rate compared to the benchmark vector group (Fig. EV2).

We assessed heart function of mice by echocardiography (Fig. 4A–C). Compared to healthy carriers, the benchmark vector group showed compromised ejection fraction, fractional shortening, and stroke volume, whereas mice that received the 2nd-generation vector did not show any heart function abnormalities. The heart weight to body weight ratio was increased in the benchmark vector-treated mice ($6.7 \pm 1.9$), but not in the 2nd-generation vector-treated mice ($4.8 \pm 0.2$), which was comparable to healthy littermates ($4.5 \pm 0.2$) (Fig. 4A). Premature ventricular contractions, enlarged ventricles, and thinning of the cardiac walls were also observed in the benchmark vector-treated animals, but not in the 2nd-generation vector group (Figs. 4B,C and EV3).

We next sought to assess whether the 2nd-generation vector led to elevated aminotransferase levels in treated animals. We measured serum alanine aminotransferase (ALT) and aspartate aminotransferase (AST) levels, which are biomarkers of liver damage (Fig. 4D). On Day 30, we detected significant increases in ALT and AST levels in the benchmark vector-treated mice (150.4 U/L and 290.8 U/L, respectively) compared to the 2nd-generation-treated mice (35.2 U/L and 93.3 U/L, respectively, $p < 0.01$ for both comparisons). Healthy carriers and 2nd-generation-treated mice has levels for both AST and ALT that statistically indistinguishable from each other ($p > 0.96$ for both comparisons).

We examined liver SMN protein expression at earlier timepoints Days 3, 8, 12, 30 and 90 post-injections (Fig. 4E). Notably, Western blot analysis revealed that the benchmark vector-treated mice exhibited strong SMN expression which was 64-fold above of healthy carrier, beginning on Day 3 followed by a rapid decline after Day 12. In contrast, SMN expression in the 2nd-generation-treated mice did not exhibit an initial peak on Day 3 and had ~6-fold less SMN expression compared to the benchmark-treated group on Day 3 (Fig. 4F). Taken together, these data demonstrated that the benchmark vector was unable to rescue heart and respiratory function in SMA mice and had indications of liver toxicity. In contrast, the 2nd-generation vector-treated mice showed no liver abnormalities and exhibited heart and respiratory functions comparable to healthy carriers.

 

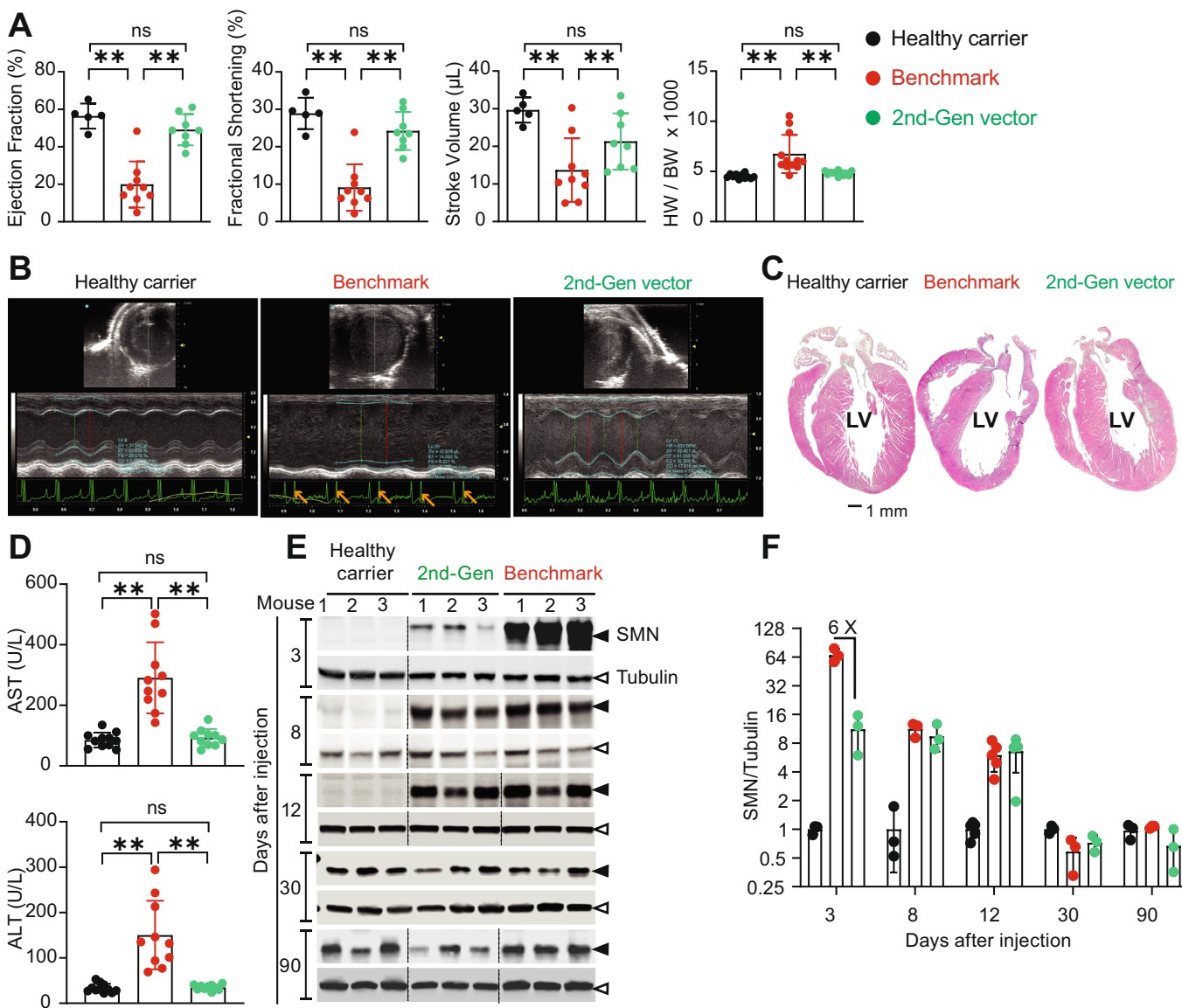

**Figure 4. The 2nd-generation vector improved cardiac and respiratory function without liver toxicity.**

(A) Echocardiographic assessment of P0-treated SMA mice, using a dose of 3.3E14 vg/kg of the benchmark ($n = 9$) or 2nd-generation vector ($n = 8$), 30 days post-administration. Age-matched healthy carriers were used as controls ($n = 5$). (B) M-mode tracings of representative transthoracic echocardiograms of the left ventricle. The yellow arrowheads indicate premature ventricular contractions. (C) H&E staining of mouse hearts showing enlargement of the left ventricle in BMK-treated mice. (D) Serum aspartate aminotransferase (AST), and alanine aminotransferase (ALT) levels were measured on Day 30, as biomarkers of liver toxicity ($n = 10$ in both the benchmark and 2nd-generation vector groups; $n = 11$ in the healthy carrier group). (E) Western blot analysis of SMN protein in mouse liver on Days 3, 8, 12, 30, and 90 post-injections of either the benchmark or 2nd-generation vector. (F) Quantitative analysis of (E). Band intensities were averaged, expressed as a ratio to β-tubulin, and normalized to healthy carriers. Age-matched healthy carriers were used as a reference. Untreated SMA mice were harvested at two weeks of age as a control. $n = 3$ animals. Bars represent the mean and error bars represent SD. One-way ANOVA. \*\*$P < 0.01$; ns, not significant. Source data are available online for this figure.

## The 2nd-generation vector greatly reduced peripheral tissue disease manifestations in SMA mice compared to the benchmark vector

SMN is normally expressed in many tissues at varying levels, suggesting that tissues that express high levels of SMN, e.g., motor neurons, are more susceptible to SMN deficiency than those requiring lower levels. In disease states where SMN is expressed in trace amounts or is absent, such as in the mouse model used here

and in the most severely affected SMA patients with congenital presentations, SMA manifests as more than a motor neuron disease (Hamilton and Gillingwater, 2013; Nash et al, 2016). While it is still unknown whether motor neurons are selectively vulnerable to SMN protein deficiency, emerging evidence indicates that SMN plays a critical role in capillary formation in peripheral tissues (Besse et al, 2020; Hua et al, 2011; Sumner and Crawford, 2018). Using the same SMA mouse model, multiple groups have shown that systemically delivered AAV9-*SMN* vectors, including the benchmark vector,

   

prolong survival and reduce body weight loss. However, peripheral disease manifestations may continue to be observed after treatment, e.g., respiratory distress, heart enlargement, eyelid rim inflammation, tail necrosis, and necrotic pinna (Dominguez et al, 2011; Foust et al, 2010; Valori et al, 2010).

In mice treated with the benchmark vector at a dose of 3.3E + 14 vg/kg, we observed necrotic pinna and watery eyes appearing at ~Day 40 that worsened over time (Fig. 5A,B). After two months, 78.0% (7/9) and 33.0% (3/9) of surviving mice exhibited necrotic pinna and watery eyes, respectively. Furthermore, by Day 90, all animals treated with the benchmark vector showed ear necrosis. With the 2nd-generation vector treatment, no animals developed ear necrosis or watery eyes throughout the 90-day study period. We observed diarrhea in 50% (10/20) of benchmark vector-treated mice on Days 7–20 versus less than 7% (1/16) in 2nd-generation vector-treated mice (Fig. 5C). This diarrhea phenotype disappeared after Day 20 in both the benchmark and 2nd-generation vector-treated groups. During necropsy of sacrificed animals from the benchmark vector group, some animals had a large amount of food lodged in the intestinal tract, especially in the cecum. We also observed enlarged hearts in some animals. We did not observe significant body weight loss in the benchmark vector-treated mice prior to death. Heart, lung, liver, and/or gastrointestinal system dysfunction might have contributed to the sudden deaths observed in the benchmark-treated group. Following a 5-month assessment period, a small cohort of 2nd-generation-treated mice ($n = 5$) appeared healthy except for a smaller body size and shortened tail (Fig. 5D and Video EV2). This cohort of mice also did not exhibit any hind-limb clasping behavior, which is an indicator of DRG toxicity and severe neurodegeneration (Van Alstyne et al, 2021). In summary, intravenous delivery of the 2nd-generation vector resulted in a reduction in peripheral disease manifestations associated with the severe SMA phenotypes, compared to the benchmark vector.

## The 2nd-generation vector restored the SMN expression profile in SMA mice back to that of healthy carrier mice by Day 90

The efficient rescue of the SMA disease phenotype achieved by the 2nd-generation vector could be a result of restoring SMN expression to physiological levels in multiple organs. To test this hypothesis, we collected major organs from SMA mice treated with 3.3E + 14 vg/kg of each vector on Days 30 and 90 and evaluated SMN expression by Western blot analysis (Fig. 6A,B). Compared to healthy carriers on Day 30, mice treated with the benchmark vector had significantly lower SMN expression in the CNS (brain and spinal cord; $p < 0.01$) and higher expression in the heart (69-fold) and quadricep muscles (35-fold). In contrast, the 2nd-generation vector-treated mice showed comparable SMN expression in brain and quadriceps, ~0.6-fold expression in the spinal cord and 36-fold expression in the heart. Compared to the benchmark vector, the 2nd-generation vector resulted in 1.8-fold and 3.5-fold more SMN in the spinal cord and brain, respectively (Fig. 6A), whereas lower levels of SMN were observed in the heart (0.5-fold) and quadriceps muscle (0.07-fold) (Fig. 6A). On Day 90, the 2nd-generation vector restored SMN expression in the brain, spinal cord, heart, and quadriceps of SMA mice to levels like healthy carrier mice ($p = 0.706$, $p = 0.219$, $p = 0.086$, $p = 0.515$, respectively) (Fig. 6B).

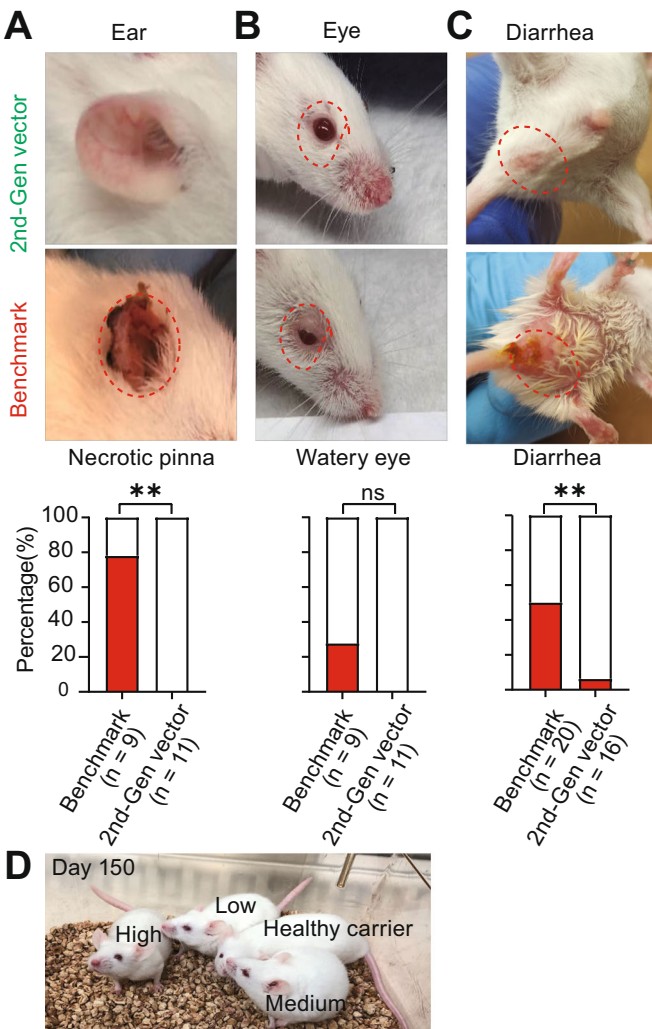

**Figure 5. The 2nd-generation vector reduced disease manifestations in peripheral tissues.**

(A–C) The frequency of necrotic pinna, watery eyes, and diarrhea in the benchmark vector or 2nd-generation vector-treated SMA mice throughout the 90-day study period. (D) Photograph of P0-treated SMA mice that received the 2nd-generation vector at different doses and healthy carrier mice on Day 150. Fischer's exact test, **$P < 0.01$; ns, not significant. Source data are available online for this figure.

In contrast, the benchmark vector expressed less SMN protein in the CNS (0.67-fold in brain; 0.68-fold in spinal cord) and more SMN protein in peripheral tissues (quadricep and heart; $p < 0.01$).

We examined the transduction efficiency of these vectors in different cell types (motor neurons, all mature neurons, microglia, and astrocytes) in the spinal cords of treated animals by immunofluorescence staining. Endogenous SMN protein was mainly distributed in motor neurons and mature neurons in healthy carrier animals, as indicated by ChAT and NeuN co-localization (Fig. 6C,D). Less SMN staining was detected in microglia and astrocytes (Fig. EV4A,B). SMA mice treated with the 2nd-generation vector showed an SMN expression pattern that was similar to that of heathy carrier animals. In contrast, the

 

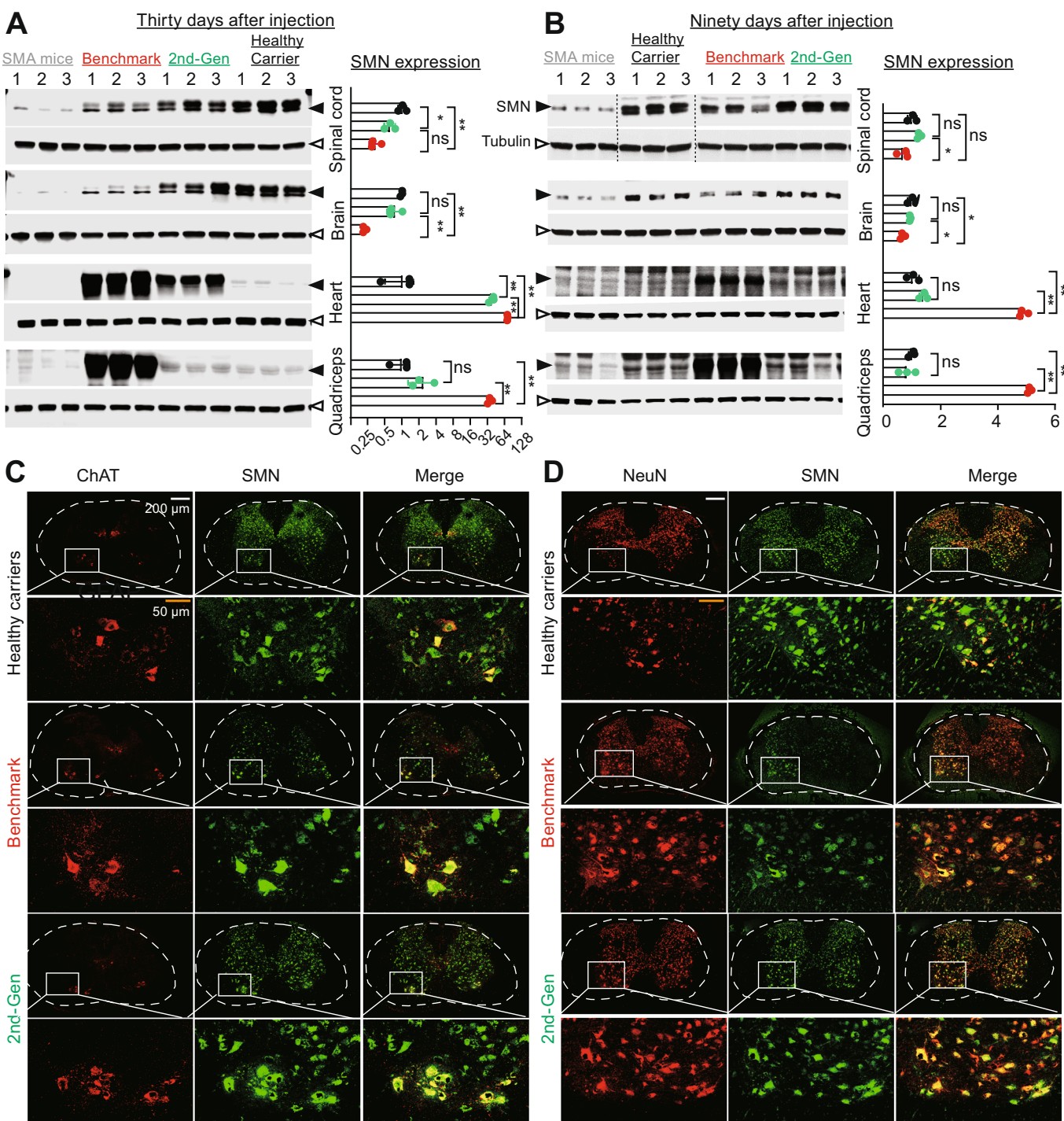

**Figure 6.   The 2nd-generation vector was preferentially expressed in the CNS compared to peripheral tissues.**

(**A, B**) SMN protein expression analysis. SMA neonates were injected with vectors via the facial vein with a dose of 3.3E + 14 vg/kg on P0. On (**A**) Day 30 and (**B**) Day 90, mouse brain, spinal cord, heart, and quadricep muscles were harvested for Western blot analysis. Age-matched healthy carriers were used as a reference. Untreated SMA mice were harvested at two weeks of age as a control. Quantitative analysis is on the right side. SMN expression shown as a ratio to β-tubulin and normalized to healthy carriers. (**C**) Immunostaining analysis in mouse lumbar spinal cord at 30 days. ChAT (red, a motor neuron marker), SMN (green), and merged (yellow). (**D**) Immunostaining of mouse lumbar spinal cord at 30 days. NeuN (red, a neuronal marker), SMN (green), and merged (yellow). Data represent the mean ± SD. One-way ANOVA, *$P < 0.05$; **$P < 0.01$; ns, not significant. Source data are available online for this figure.

   

benchmark vector-treated animals showed less SMN expression in motor neurons and pan-neurons.

## Discussion

Three widely approved drugs, nusinersen, risdiplam, and onasemnogene abeparvovec, have rapidly changed the SMA therapy landscape in recent years (Baranello et al, 2021; Darras et al, 2021; Finkel et al, 2017; Mendell et al, 2017; Mercuri et al, 2018). The implementation of newborn screening has also enabled early therapeutic intervention, leading to better outcomes for patients (Davidson and Farrar, 2022; Hordeaux et al, 2020b; Strauss et al, 2022).

At the time of this study, more than 3,000 patients have been treated with intravenously administered onasemnogene abeparvovec, which has led to prolonged survival, attainment of motor milestones, and enhanced quality of life (Mendell et al, 2017; Strauss et al, 2022). Despite this remarkable progress, there remain unmet clinical needs and associated adverse reactions that suggest opportunities to further optimize gene therapy for SMA. Uncontrolled high-level expression of SMN under the strong *CMVen/CB* promoter and delivery by AAV9 could be detrimental to cells or organs that cannot tolerate abundant SMN protein. In addition, high doses of AAV9 capsid may trigger immune responses and liver damage. The US prescribing label lists the most common adverse events reported in clinical trials as elevated transaminases (27.8%) and vomiting (6.7%). Furthermore, a meta-analysis of clinical safety data conducted by Novartis on onasemnogene abeparvovec through 2019, liver-associated adverse events were reported in 34% of 100 patients across five clinical trials and in 23% of 43 patients in a managed access program and diseases registry (Chand et al, 2021). To date, two patients have died from acute liver failure (NOVARTIS 2022b). The warnings and precautions section of the US prescribing label cites the potential for a systemic immune response, thrombocytopenia, thrombotic microangiopathy, and elevated troponin-I. The label also includes a black box warning of cases of acute serious liver injury with elevated transaminases and acute liver failure with fatal outcomes. Our data in SMA mice suggest that excessive hepatic SMN expression may be a cause of liver toxicity and even early death (Figs. 1C,E–G and 4E).

Upon reducing SMN expression in the liver by incorporating liver-specific miRNA binding sites in the vector genome, or by switching the *CMVen/CB* promoter to a regulatory cassette derived from the endogenous *SMN1* promoter that has low activity in liver (Fig. 4F), abolished liver toxicity (Figs. 1H–K and 4E). We also observed supraphysiological expression of SMN in the skeletal muscles and the heart of SMA mice receiving the benchmark vector, which may lead to adverse effects, including cardiac and respiration dysfunction.

Finally, because of DRG toxicities observed in pre-clinical studies (NHP and pigs) with rAAV9-*SMN1* (Hinderer et al, 2018; Hordeaux et al, 2020a), the FDA placed a temporary partial clinical hold on the clinical trial for intrathecal onasemnogene abeparvovec administration. Subsequent investigations showed that DRG toxicity is dependent upon transgene expression and AAV dose (Buss et al, 2022; Hordeaux et al, 2020b; Palazzi et al, 2022; Tukov et al, 2022). These findings necessitate monitoring for potential organ toxicity in patients treated with onasemnogene abeparvovec and other AAV gene therapies. In our hands, SMA mice treated with the 2nd-generation vector and observed

for 150 days post-administration did not exhibit hind-limb clasping behavior, a neurological disorder attributed to DRG toxicity and sever neurodegeneration (Taylor et al, 2001; Van Alstyne et al, 2021).

The long-term impact of not treating peripheral tissues in SMA, or of supraphysiological expression of SMN is unknown. SMN protein is expressed at varying levels in many organs and is theoretically functional in all cells (Hamilton and Gillingwater, 2013; Iascone et al, 2015; Mercuri et al, 2022; Singh et al, 2017). While all patients with SMA exhibit motor neuron degeneration and manifest neurogenic muscle weakness, only a few with the most severe disease presentation (SMA type 0 and some with SMA type 1) who produce a trace amount or no SMN, have exhibited systemic disease, e.g., congenital heart malformations, bradycardia, altered glucose regulation, and distal digital necrosis. In severe SMA mouse models that express little to no SMN, there is evidence for a microvasculopathy involving defective angiogenesis and blood vessel formation that results in systemic manifestations (Hamilton and Gillingwater, 2013; Nash et al, 2016; Wirth, 2021; Wirth et al, 2020).

In this study, we developed a 2nd-generation scAAV9 vector expressing a *co-hSMN1* coding sequence under the control of a regulatory cassette derived from the endogenous human *SMN1* promoter. A head-to-head comparison demonstrated therapeutic advantages of the 2nd-generation vector over a benchmark vector, whose vector genome is identical to onasemnogene abeparvovec, as evidenced by prolongation of life span and improved heart, respiratory, and motor function. The 2nd-generation vector also conferred efficient correction of peripheral tissue disease manifestations, eliminated liver toxicity, and had a broader therapeutic time window for clinical benefit. Our results suggest that the improved efficacy and safety of the 2nd-generation vector are attributable to the restoration of SMN expression closer to physiological levels in multiple organs, including both CNS and non-CNS tissues, following intravenous administration. In contrast, the benchmark vector expressed less SMN in the CNS and supraphysiological levels of SMN in peripheral tissues. At 1/3 of the dose the 2nd-generation vector resulted in a longer lifespan in SMA mice than the benchmark vector (Fig. 2C–F) and had a better safety profile.

The overall improvement in efficacy with the 2nd-generation vector may be due to the *SMN1* promoter, which enables exogenous SMN expression that is similar to the endogenous SMN expression profile throughout the body. Based on the results reported here, the 2nd-generation vector offers substantial therapeutic benefits over onasemnogene abeparvovec in SMA mice and potentially in SMA patients. Notably, performing clinical trials with the 2nd-generation vector can be challenging in Type-1 SMA patients where the birth screening and onasemnogene abeparvovec are available. The successful use of a regulatory cassette derived from the endogenous SMN1 promoter also suggests that endogenous gene-specific promoters for gene replacement or gene silencing holds promise for safer and more efficacious AAV gene therapy.

## Methods

### Mice

Animal experiments (PROTO202000038) were approved by the Institutional Animal Care and Use Committee (IACUC) of UMASS

 

Chan Medical school. SMA mice (FVB.SMNΔ7; SMN2; Smn-, IMSR_JAX:005025), breeding pairs were purchased from Jackson Laboratory. SMA parental mice were mated in the UMass Chan Medical School Animal Facility. Mice were housed in standard cages in a temperature-controlled room (22–24 °C) with a 12 h dark–light cycle and fed with standard chow (LabDiet, #5P7622;.22.5% protein, 5.4% fat, 4% fiber, 50% polysaccharide). Tail samples were collected from the pups upon delivery, and DNA was extracted using KAPA lysis buffer (KAPA Express Extract, Roche, 07961626001) from each sample. Genotyping was performed as previously published (Foust et al, 2010). Litter was culled to three pups per cage after genotyping. SMA or healthy control animals were injected with different doses of AAV vectors or an empty vector control in a total volume of 50 μL PBS buffer via the facial vein, as previously described (Gessler et al, 2019). SMA mice were randomly divided into different groups as described in the figure legends and age-matched healthy carriers were used as a control. Both male and female mice were used in the study. The study was not conducted in a double-blinded way. Before tissue collection, animals were perfused using a mini perfusion pump under deep anesthesia. Animals were perfused with PBS and tissues were snap-frozen with liquid nitrogen and kept in −80 °C for Western blot and DNA/RNA analysis. For immunostaining, animals were perfused with PBS saline and 4% paraformaldehyde solution. Tissues were collected and transferred into 4% paraformaldehyde solution to fix for 24 h. Afterward, samples were moved to 30% sucrose overnight to dehydrate. Tissues were then embedded, and slides with sections were prepared for staining.

## H&E staining and pathological analysis of liver samples

Animals were perfused with PBS under deep anesthesia. For each animal, at least two samples from distinct lobules of the liver were collected and fixed immediately in 10% formaldehyde for 24 h. Samples were dehydrated and embedded in paraffin. 6 μm sections were cut from each tissue block and stained with hematoxylin and eosin (H&E). Images were captured using a Leica Thunder DMi8 microscope system at two magnification levels. Lymphocyte infiltration and spotty or widespread necrosis with architectural disturbance or collapse in any area of the liver samples was defined as liver damage.

## Cell culture

Neuro2a cells were purchased from ATCC (CCL-131) and maintained in Eagle's Minimum Essential Medium (EMEM, ATCC 30-2003) supplemented with 10% fetal bovine serum (FBS, HyClone). Cells were cultured in a humidified 37 °C/5% $CO_2$ incubator. For in vitro expression assays with different *SMN* constructs, cells were seeded 18–24 h before transfection at a density of $5 \times 10^5$ cells/well in a 6-well plate. Cells were transfected with Lipofectamine 2000 (Thermo Fisher Scientific) in accordance with the manufacturer's instructions. Cells were harvested 48 hr post-transfection and protein samples were prepared for Western blot analysis.

## Vector design, construction, and production

The benchmark construct was created by incorporating a synthesized DNA fragment containing the chicken beta-actin (*CB*) promoter with the cytomegalovirus (*CMV*) immediate early enhancer (*CB/CMV*), SV40 intron, wild-type *SMN1* coding sequence, and bovine growth hormone polyadenylation (*bgh-polyA*) signal sequence. This construct is identical in design to the AVXS-101 vector used in pre-clinical studies. The construct was incorporated into a self-complementary AAV (scAAV) vector. The codon-optimized human *SMN1* coding sequence was incorporated into the scAAV-*CB/CMV* constructs without an intron, or with a synthetic intron. For the 2nd-generation vector, the *SMN1* endogenous promoter was adopted from a previously described sequence (Echaniz-Laguna et al, 1999). Specifically, the SMN promoter (−767 to +152) sequence was synthesized by Integrated DNA Technologies (IDT) and incorporated upstream of the codon-optimized *SMN1* coding sequence in a scAAV genome. rAAV was produced by transient HEK293 cell transfection and CsCl ultracentrifugation by the UMass Chan Medical School Viral Vector Core, as previously described (Sena-Esteves and Gao, 2020). Vector preparations (vg/mL) were titered by ddPCR, and the purity was assessed by 4–12% SDS-polyacrylamide gel electrophoresis and Flamingo staining (Invitrogen).

## Detect genome copy from DNA-extracted animal tissues

Frozen samples were ground into powder in liquid nitrogen. DNA was isolated from tissue powder using a tissue DNA isolation kit (QIAGEN, Cat#: 69504). DNA concentration and quality was measured by using a nanodrop. 10 ng of each extracted DNA sample were added to PCR reactions for the transgene (SMN1) copy number assay (Forward primer: 5′-CACCCGCGGGTTTGC TATG, Reverse primer: 5′-TCATCGCTCTGGCCTGT, Probe: FAM-5′-CCACTGCCGCCGCTGCTCAT); VIC labeled Tfrc (Thermo Scientific, Cat#: 4458366) was used as a loading control, and ddPCR supermix reagent (BioRad, Cat#: 1863024) was used for the reaction buffer. ddPCR droplets were generated in the QX100/200 Droplet Generator using BioRad Droplet Generation Oil following the manufacturer's instructions. After the droplets were formed, they were carefully transferred to a PCR machine for amplification under the following conditions: initial denaturation at 95 °C for 10 min, followed by 95 °C for 30 s; 60 °C for 1 min and 98 °C for 10 min, for 40 cycles. The ddPCR plate was read in a QX200 reader, and the results were analyzed using the Quanta-SoftTM software.

## Rotarod test

Rotarod testing was conducted on Days 30 and 90 post-injections. Animals were gently placed on top of the rod in a rotarod device and allowed to explore the rotarod for several minutes without rotating. The rod was then slowly rotated with accelerating speed. Training was done 5–10 times until the mice were familiarized with the rolling rod. Each mouse was subjected to two consecutive days of training before testing. The time (s) that each animal stayed on the rolling rod without falling was measured and recorded. Each mouse performed three tests, and the best performance was recorded.

## Surface righting test

Surface righting tests were performed on Days 3, 5, 7, 9, 11, and 13. When performing the test, pups were placed on a cotton sheet in a

 

supine position and released. The time it took for each pup to return to a prone position (with all four paws on the ground) was recorded. A total of 30 s was allowed for each test, the test was repeated a total of three times, and the results from the trials were averaged (Feather-Schussler and Ferguson, 2016).

## Grid test

The grid test was conducted on Days 9, 10, and 11. Our method was derived and modified from a previously published procedure (Tillerson and Miller, 2003). The grid apparatus consisted of a 30 degree-angled grid mesh. The grid mesh was constructed 20 cm above a soft surface. Mice were slowly placed in the center of the angled grid and held until they stably grabbed the grid with their paws. Mice were released and the time it took for the mice to drop onto the soft surface was measured. A maximum of 60 s was allowed for each test, the test was repeated a total of three times, and the results from the trials were averaged.

## Echocardiography

Animals were treated with different vectors at P0. On Day 30, echocardiograms were conducted to test heart function. The abdomen and chest of each animal were treated with a depilatory cream. Animals were induced with 2.0% isoflurane mixed with 0.5 L/min of 100% $O_2$ and gently affixed to a heated physiologic platform of the Vevo 3100 imaging system (Visual Sonics, Toronto, ON, Canada). Electrode cream was applied to each limb. Body temperature was continually monitored and maintained at 37 °C with a rectal temperature probe. Isoflurane was administered by nose cone and the concentration reduced to 1.0% isoflurane mixed with 0.5 L/min of 100% $O_2$. All animals were imaged at a heart rate of at least 450–500 beats per minute. 2D and M-mode images were obtained in the parasternal long and short axes with a 50 MHz transducer (MX550S) as previously described (Respress and Wehrens, 2010; Scherrer-Crosbie and Thibault, 2008). Imaging analysis was performed offline using Vevo LAB image analysis software. LV volumes were derived from M-mode measurements using the following formulas: LV diastolic volume = [(7.0/(2.4 + LVIDd)]*LVIDd$^3$ and LV systolic volume = [(7.0/(2.4 + LVIDs)] *LVIDs$^3$. All imaging was done in the UMass Chan Medical School Cardiovascular & Surgical Models Core. LV; left ventricle, LVIDd; left ventricular internal diameter in diastole, LVIDs; left ventricular internal diameter in systole.

## Immunostaining of transversus abdominis (TVA) muscle

Isolation of TVA muscle was followed using a protocol from Lyndsay (Murray et al, 2014). Neuromuscular junction (NMJ) staining was adopted from the same protocol with some modifications. Briefly, TVA muscle was placed into a 24-well plate and permeabilized by adding 500 μL/well of 2% Triton X-100 in PBS solution and rocking the plate for 30 min at 100 rpm. Blocking solution (10% goat serum, 1% Triton X-100 in PBS) was then added for 30 min at room temperature. The primary antibody (rabbit anti-neurofilament, Abcam, Catalog # ab254348) was diluted in blocking buffer at 1:100 and incubated for 48 h at 4 °C. After three washes with PBS + 0.05% Tween 20, the secondary antibody (Invitrogen Goat anti-Rabbit Alexa 488, Catalog # A32731, 1:500

dilution) and α-bungarotoxin (Alexa Fluor™ 594 conjugate, Thermo Fisher, Catalog # B13423, 1:500 dilution) were incubated for 4 h at room temperature. After three washes with PBS + 0.05% Tween 20, TVA muscles were mounted onto glass slides. Images were taken with a Leica SP8 confocal microscope system.

## Immunostaining of spinal cord

Animals were perfused with PBS to flush out the blood followed by a 4% formaldehyde solution to fix the tissues. The vertebral column was isolated and further fixed by immersion in a 4% formaldehyde solution overnight. A 30% sucrose buffer was used afterward to dehydrate the tissue for 24 h. Lumbar spine (L1-L5) was dissected, mounted in OCT embedding compound, and frozen at −80 °C. Cryosections were cut at a thickness of 15 μm around the lumbar spine L2 level and mounted onto gelatin-coated histological slides. Tissue slides were prepared in an antigen retrieval process using citrate buffer (pH 6, Agilent Technologies, Catalog #: S236984-2) and heating at 100 °C for 5 min. Slides were permeabilized in 0.5% Triton X-100 in PBS for 30 min and incubated in blocking buffer (10% goat serum in PBS) for 30 min at room temperature. Primary antibodies were diluted (anti-SMN, 1:500, Cell Signaling Technology, Catalog # 12976; anti-choline acetyltransferase, 1:400, Sigma-Aldrich, Catalog # HPA048547; and anti-NeuN, 1:400, Thermo Fischer Scientific, Catalog # PA5-78499) in blocking buffer and added onto the slides overnight at 4 °C. Slides were washed with 0.5% Triton X-100 in PBS three times for 15 min each. Fluorescent-conjugated secondary antibodies (Goat anti-Rabbit IgG Secondary Antibody, Alexa Fluor® 488, 1:500, Invitrogen, Catlog#: A32731; Goat anti-Mouse IgG Secondary Antibody, Alexa Fluor® 488, 1:1000, Fisher Scientific, Catlog#: A11029; Goat anti-Rabbit IgG Secondary Antibody, Alexa Fluor® 594, 1:1000, Fisher Scientific, Catlog#: A32740) were applied and incubated for 1 hr at room temperature. After three washes the slides were mounted with anti-fade media. Fluorescence images were visualized with a Leica SP8 laser confocal microscope.

## Western blot analysis

All tissue samples were snap-frozen in liquid nitrogen after harvesting. For extracting protein, frozen samples were ground into powder in liquid nitrogen. Protein was prepared from tissue powder using T-PER™ Tissue Protein Extraction Reagent (Thermo Scientific, Catalog # 78510) following the manufacturer's instructions. Protein concentration was determined by using the bicinchoninic acid method (BCA, Thermo Fisher Scientific). 20 μg of total protein per sample was loaded onto a 4–20% gradient gel (Bio-Rad), and proteins were size-separated and transferred onto a Millipore Immobilon-NC membrane. The membranes were placed in Intercept (PBS) blocking buffer (Li-Cor, Catalog # 927-70001) for 1 h at room temperature. Mouse anti-SMN antibody (Cell Signaling Technology, Catalog # 12976) was diluted at a dilution factor of 1:1000 in Intercept T20 (PBS) antibody diluent (Li-Cor, Catalog # 927-75001) and incubated with the membranes overnight at 4 °C. The membranes were washed three times for 5 min with PBS + 0.05% Tween 20 and then incubated with goat anti-mouse IgG polyclonal antibody, IRDye 800CW (Li-Cor, Catalog # 926-32210) secondary antibodies in antibody diluent at a dilution factor of 1:5000 for 1.5 h at room

## The paper explained

### Problem

Spinal muscular atrophy (SMA) is the most common genetic cause of death in childhood. The disease is caused by loss of function mutations in the survival motor neuron 1 gene (SMN1) encoding SMN. Onasemnogene abeparvovec (Zolgensma®) is an approved adeno-associated virus 9 (AAV9) vector ubiquitously expressing a human SMN1 cDNA transgene for infants with SMA, however, lack of efficacy and adverse events have been observed in patients following treatment.

### Results

We developed a second-generation AAV9 gene therapy expressing a codon-optimized hSMN1 transgene driven by a promoter derived from the native hSMN1 gene. This vector restored SMN expression close to physiological levels in the central nervous system and major systemic organs of a severe SMA mouse model. In a head-to-head comparison between the second-generation vector and a benchmark vector, identical in design to onasemnogene abeparvovec, the second-generation vector showed better safety and improved efficacy.

### Impact

The second-generation vector offers substantial therapeutic benefits over onasemnogene abeparvovec in SMA mice and potentially in SMA patients. The successful use of a regulatory cassette derived from the endogenous SMN1 promoter suggests that endogenous gene-specific promoters for gene replacement or gene silencing holds promise for safer and more efficacious AAV gene therapy.

temperature. The membranes were washed three times for 5 min in PBS + 0.05% Tween 20, and images were taken in an Odyssey CLX scanning system (Li-Cor). Quantification analysis was done using ImageJ open-source software.

## Statistical analysis

The data were analyzed by using Graph Pad Prism software (Version 9.3.1). All results are shown as the mean ± standard deviation unless otherwise specified. The unpaired two-sided student's t-test was performed to compare two groups. One-way ANOVA analysis was used to compare more than two groups. Fisher's exact test was used to compare the frequencies of peripheral tissue disease manifestations. $P < 0.05$ was considered significant for all statistical analyses.

For more information: https://www.zolgensma.com/how-zolgensma-works and https://www.curesma.org/approaches-to-drug-development/.

## Data availability

This study includes no data deposited in external repositories.

## Peer review information

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

## Acknowledgements

This research was sponsored by CANbridge Pharmaceuticals.

## Author contributions

**Qing Xie**: Conceptualization; Resources; Data curation; Formal analysis; Validation; Investigation; Methodology; Writing—original draft; Writing—review and editing. **Xiupeng Chen**: Data curation; Formal analysis; Validation; Investigation; Methodology. **Hong Ma**: Resources; Data curation; Investigation. **Yunxiang Zhu**: Investigation; Methodology. **Yijie Ma**: Data curation; Investigation; Methodology. **Leila Jalinous**: Data curation; Investigation. **Gerald F Cox**: Methodology; Writing—original draft. **Fiona Weaver**: Data curation; Investigation; Methodology; Writing—original draft. **Jun Yang**: Data curation; Formal analysis; Writing—review and editing. **Zachary Kennedy**: Data curation; Writing—original draft. **Alisha Gruntman**: Investigation. **Ailing Du**: Investigation. **Qin Su**: Resources. **Ran He**: Resources. **Phillip WL Tai**: Investigation; Methodology; Writing—review and editing. **Guangping Gao**: Formal analysis; Supervision; Funding acquisition; Writing—original draft; Project administration; Writing—review and editing. **Jun Xie**: Conceptualization; Data curation; Formal analysis; Supervision; Funding acquisition; Methodology; Writing—original draft; Project administration; Writing—review and editing.

## Disclosure and competing interests statement

QX, HM, GG, and JX are inventors of a patent application filed by the University of Massachusetts Chan Medical School on SMA gene therapy. GG is a scientific co-founder of Voyager Therapeutics, Adrenas Therapeutics, AAVAA Therapeutics, and Aspa Therapeutics and holds equity in these companies. PT, GG, and JX are inventors of patents related to AAV-based gene therapy, some of which were licensed to commercial entities. YM, LJ, FW, JY, and ZK are employees of CANbridge Pharmaceuticals. YM, LJ, GC, JY, and ZK hold equity and/or stock options in CANbridge Pharmaceuticals. The other authors declare no competing interests.

# Expanded View Figures

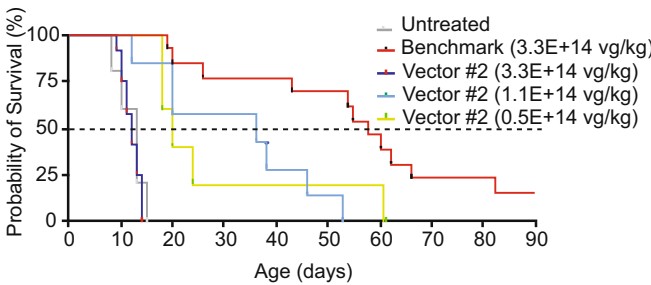

**Figure EV1. Survival of SMA mice injected at P0 with Vector 2 at three different doses.**

Non-injected SMA mice were used as a control.

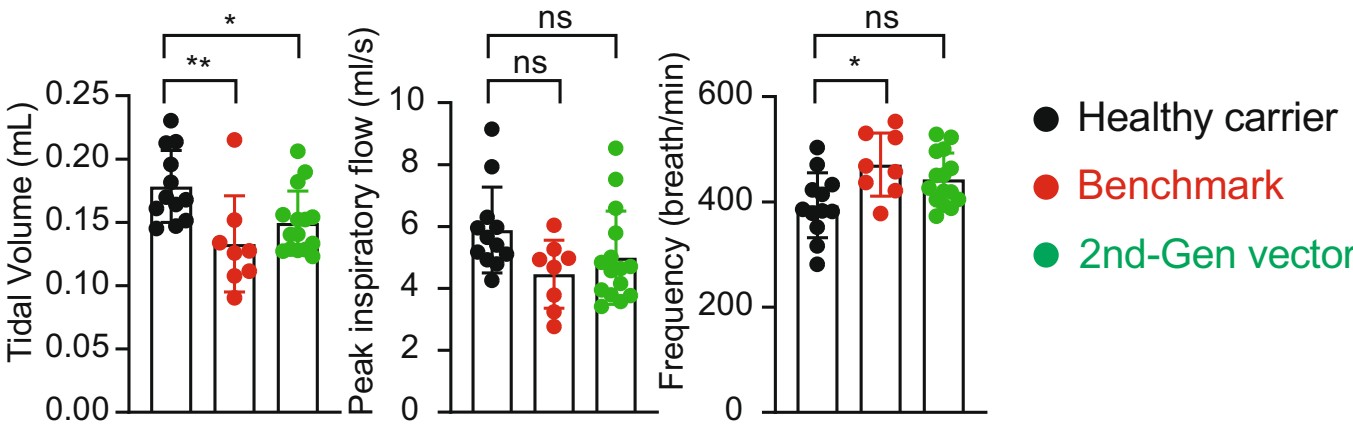

**Figure EV2. Respiratory function assessment by plethysmography in SMA-treated mice and healthy carriers.**

On Day 30, respiratory function was assessed by tidal volume, peak inspiratory flow, and respiratory rate. Bars represent the mean and error bars represent SD. One-way ANOVA, *$P < 0.05$; **$P < 0.01$; ns, not significant.

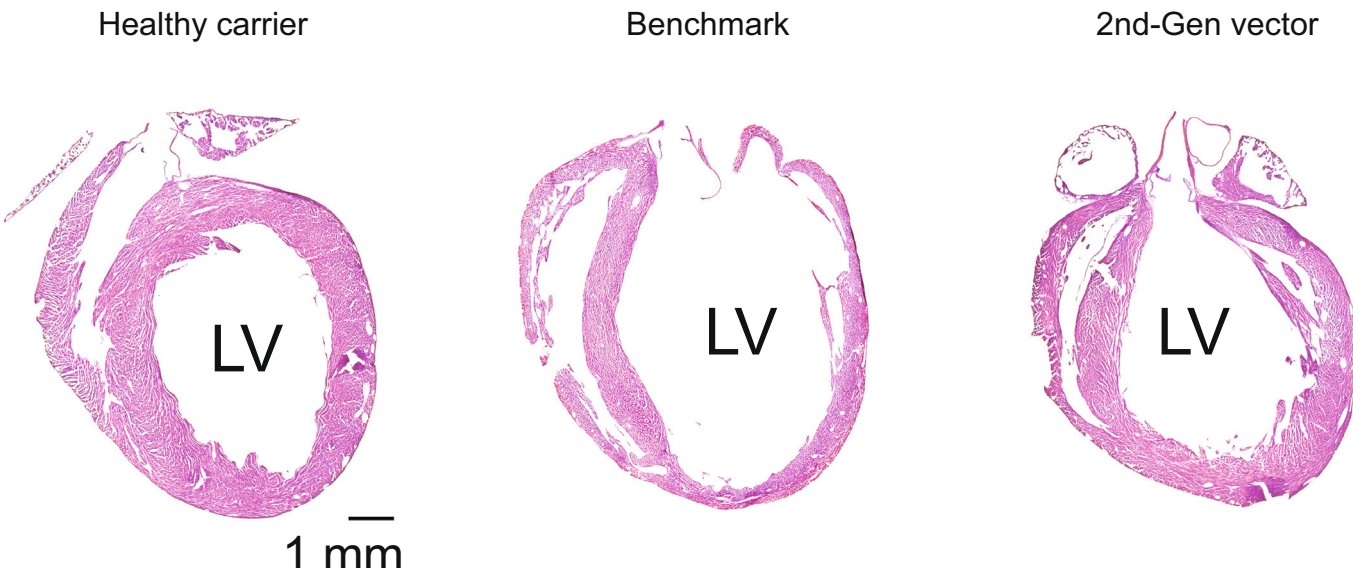

Healthy carrier Benchmark 2nd-Gen vector

1 mm

**Figure EV3.  Heart H&E staining on Day 90 of SMA injected at P0 with the benchmark and 2nd-generation vectors.**

Healthy littermates were used as a control.

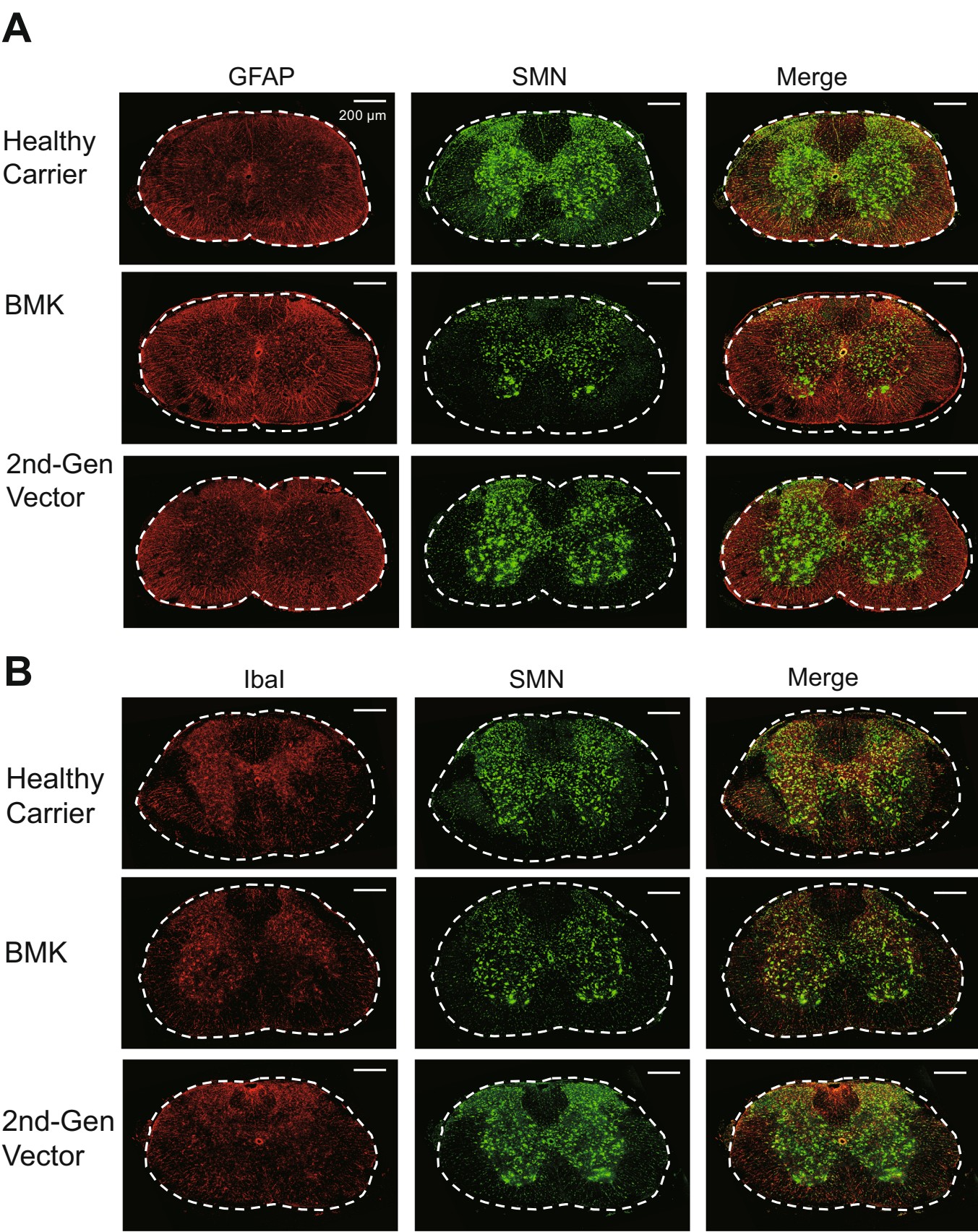

◀ **Figure EV4. Immunostaining of mouse lumbar spinal cord on Day 30.**

(A) GFAP (red, an astrocyte marker), SMN (green), and merged (yellow). (B) Immunostaining of mouse lumbar spinal cord on Day 30. Iba1 (red, a microglial marker), SMN (green) and merged (yellow).

 