## [Peer Review File · EMBO Molecular Medicine]

Improved gene therapy for spinal muscular atrophy in mice using codon-optimized hSMN1 transgene and hSMN1 gene-derived promotor

Jun Xie, Qing Xie, Xiupeng Chen, Hong Ma, Yuxiang Zhu, Yijie Ma, Leila Jalinous, Gerald Cox, Fiona Weaver, Jun Yang, Zachary Kennedy, Ailing Du, Alisha Gruntman, Qin Su, Ran He, Phillip Tai, and Guangping Gao

Corresponding author(s): Jun Xie (jun.xie@umassmed.edu) , Guangping Gao (Guangping.Gao@umassmed.edu)

Review Timeline:

Submission Date:	2nd Nov 23
Editorial Decision:	7th Dec 23
Revision Received:	31st Jan 24
Accepted:	1st Feb 24

Editor: Zeljko Durdevic

Transaction Report:

7th Dec 2023

Dear Dr. Xie,

Thank you for the submission of your manuscript to EMBO Molecular Medicine and please accept my apologies for the delay in getting back to you, which is due to the fact that one referee needed more time to complete his/her review. We have now received feedback from the three reviewers who agreed to evaluate your manuscript. As you will see from their reports pasted below all three referees support publication of your manuscript raising important but minor concerns. Therefore, I am pleased to inform you that we will be able to accept your manuscript pending the following final amendments:

- 1) Please address all referee concerns. No additional experiments are required.
- 2) Reorder manuscript sections: Abstract, Introduction, The Paper Explained, Results, Discussion, Materials and Methods, Acknowledgements, Disclosure and competing interests statement, For More Information, References, Figure legends, Tables and their legends, Expanded View Figure legends
- 3) Figures:
 - Please remove all figure from the manuscript text and upload individual, high-resolution file for each main and EV figure. Suppl. Figures 2-5 should be made Figure EV 1-4 and the legends should stay in the manuscript. Suppl. Fig. 1 would best be made an appendix figure - uploaded in PDF format, with a ToC, legend removed from manuscript text and added underneath the figure, and the figure renamed "Appendix Figure S1". Please also update all figure callouts in the main text. For more information on figure presentation please check "Author Guidelines".
<https://www.embopress.org/page/journal/17574684/authorguide#datapresentationformat>
 - When presenting images of western blots from different membranes please make sure to clearly indicate it, e.g. with white dotted line, to prevent any doubt of splice sites. This needs to be consistent throughout all the western blots. For example, splice sites are not clear in Figure 4E and Figure 6B.
- 4) Author checklist: Please submit a complete checklist. <https://www.embopress.org/pb-assets/embo-site/EMBO%20Press%20Author%20Checklist-1642513524327.xlsx>
- 5) In the main manuscript file, please do the following:
 - Please address all comments suggested by our data editors listed below:
 - o DAS: Please note that the data availability statement is not provided in the manuscript.
 - o Figure legends:
 1. The legend for figures 3a-c is incorrectly labelled as 3a-b in the manuscript. This needs to be rectified.
 2. Please note that in figures 1i, k; 4a, d, f; 5a-c; there is a mismatch between the annotated p values in the figure legend and the annotated p values in the figure file that should be corrected.
 3. Please note that information related to n is missing in the legends of figures 1i, k; 3a-c; 4d; 6a-b, supplementary figure 3.
 4. Please note that the error bars are not defined in the legends of figures 1i, k.
 5. Please note that scale bar and its definition are missing for figures 4c; 6c-d, supplementary figures 4; 5a-b.
 6. Please note that in figure 1f; the scale bar unit should be corrected from μM to μm .
 - Add up to 5 keywords.
 - Remove data not shown on p.10 and 12.
 - In M&M, provide the antibody dilutions that were used for each antibody.
 - In M&M, statistical paragraph should reflect all information that you have filled in the Authors Checklist, especially regarding randomization, blinding, replication etc.
 - Add "Disclosure Statement & Competing Interests". We updated our journal's competing interests policy in January 2022 and request authors to consider both actual and perceived competing interests. Please review the policy <https://www.embopress.org/competing-interests> and update your competing interests if necessary.
 - Author contributions: Please remove it from the manuscript and specify author contributions in our submission system. CRediT has replaced the traditional author contributions section because it offers a systematic machine-readable author contributions format that allows for more effective research assessment. You are encouraged to use the free text boxes beneath each contributing author's name to add specific details on the author's contribution. More information is available in our guide to authors:
<https://www.embopress.org/page/journal/17574684/authorguide#authorshipguidelines>
 - Please add data availability statement. If no data are deposited in public repositories, add the sentence: This study includes no data deposited in external repositories.
 - Correct the reference citation in the text and reference list. In the text a reference should be cited by author and year of publication. Include a space between a word and the opening parenthesis of the reference that follows. In the reference list, citations should be listed in alphabetical order. Where there are more than 10 authors on a paper, 10 will be listed, followed by "et al.". Please check "Author Guidelines" for more information.
<https://www.embopress.org/page/journal/17574684/authorguide#referencesformat>
- 6) Movies: Rename the files to Movie EV1, etc. Remove their legends from the main manuscript file and zipp each legend as a readme.txt file with its corresponding movie.
- 7) The Paper Explained: Please provide "The Paper Explained" and add it to the main manuscript text. Please check "Author Guidelines" for more information. <https://www.embopress.org/page/journal/17574684/authorguide#researcharticleguide>

8) Synopsis: Every published paper now includes a 'Synopsis' to further enhance discoverability. Synopses are displayed on the journal webpage and are freely accessible to all readers. They include separate synopsis image and synopsis text.

- Synopsis image: Please provide a striking image or visual abstract as a high-resolution jpeg file 550 px-wide x (250-400)-px high to illustrate your article.

- Synopsis text: Please provide a short standfirst (maximum of 300 characters, including space) as well as 2-5 one sentence bullet points that summarise the paper as a .doc file. Please write the bullet points to summarise the key NEW findings. They should be designed to be complementary to the abstract - i.e. not repeat the same text. We encourage inclusion of key acronyms and quantitative information (maximum of 30 words / bullet point). Please use the passive voice.

9) For more information: This space should be used to list relevant web links for further consultation by our readers. Could you identify some relevant ones and provide such information as well? Some examples are patient associations, relevant databases, OMIM/proteins/genes links, author's websites, etc..

10) As part of the EMBO Publications transparent editorial process initiative (see our Editorial at <http://embomolmed.embopress.org/content/2/9/329>), EMBO Molecular Medicine will publish online a Review Process File (RPF) to accompany accepted manuscripts. This file will be published in conjunction with your paper and will include the anonymous referee reports, your point-by-point response and all pertinent correspondence relating to the manuscript. Let us know whether you agree with the publication of the RPF and as here, if you want to remove or not any figures from it prior to publication. Please note that the Authors checklist will be published at the end of the RPF.

11) Please provide a point-by-point letter INCLUDING my comments as well as the reviewer's reports and your detailed responses (as Word file).

I look forward to reading a new revised version of your manuscript as soon as possible.

Yours sincerely,

Zeljko Durdevic

*** Instructions to submit your revised manuscript ***

1) a .docx formatted version of the manuscript text (including Figure legends and tables)

2) Separate figure files*

3) supplemental information as Expanded View and/or Appendix. Please carefully check the authors guidelines for formatting Expanded view and Appendix figures and tables at <https://www.embopress.org/page/journal/17574684/authorguide#expandedview>

4) a letter INCLUDING the reviewer's reports and your detailed responses to their comments (as Word file).

5) The paper explained: EMBO Molecular Medicine articles are accompanied by a summary of the articles to emphasize the major findings in the paper and their medical implications for the non-specialist reader. Please provide a draft summary of your article highlighting

This may be edited to ensure that readers understand the significance and context of the research.

Please refer to any of our published articles for an example.

6) For more information: There is space at the end of each article to list relevant web links for further consultation by our readers. Could you identify some relevant ones and provide such information as well? Some examples are patient associations, relevant databases, OMIM/proteins/genes links, author's websites, etc...

7) Author contributions: the contribution of every author must be detailed in a separate section.

8) EMBO Molecular Medicine now requires a complete author checklist

(<https://www.embopress.org/page/journal/17574684/authorguide>) to be submitted with all revised manuscripts. Please use the checklist as guideline for the sort of information we need WITHIN the manuscript. The checklist should only be filled with page numbers where the information can be found. This is particularly important for animal reporting, antibody dilutions (missing) and exact values and n that should be indicated instead of a range.

9) Every published paper now includes a 'Synopsis' to further enhance discoverability. Synopses are displayed on the journal webpage and are freely accessible to all readers. They include a short stand first (maximum of 300 characters, including space) as well as 2-5 one sentence bullet points that summarise the paper. Please write the bullet points to summarise the key NEW findings. They should be designed to be complementary to the abstract - i.e. not repeat the same text. We encourage inclusion of key acronyms and quantitative information (maximum of 30 words / bullet point). Please use the passive voice. Please attach these in a separate file or send them by email, we will incorporate them accordingly.

You are also welcome to suggest a striking image or visual abstract to illustrate your article. If you do please provide a jpeg file 550 px-wide x 300-800px high.

10) A Conflict of Interest statement should be provided in the main text

11) Please note that we now mandate that all corresponding authors list an ORCID digital identifier. This takes <90 seconds to complete. We encourage all authors to supply an ORCID identifier, which will be linked to their name for unambiguous name identification.

Currently, our records indicate that the ORCID for your account is 0000-0001-9565-1567.

Link Not Available

Photos 400-800 DPI

*Additional important information regarding figures and illustrations can be found at

<https://bit.ly/EMBOPressFigurePreparationGuideline>. See also figure legend preparation guidelines:

<https://www.embopress.org/page/journal/17574684/authorguide#figureformat>

***** Reviewer's comments *****

Referee #1 (Comments on Novelty/Model System for Author):

An appropriate mouse model of SMA was used in the study.

Referee #1 (Remarks for Author):

This manuscript by Xie et al demonstrates the potential of codon-optimized SMN1 gene expression regulated by an endogenous promoter via an adeno-associated virus vector for improved gene therapy application for spinal muscular atrophy (SMA). SMA is a deadly disease with no long-term therapy until recently when recombinant adeno-associated virus vector (AAV)-based therapeutics has demonstrated efficacy. Nonetheless, vector associated immune response and unregulated transgene expression are encountered. Data provided in this manuscript indicate the potential of codon-optimized gene expression using its own promoter can overcome the limitations. The experiments have been conducted well and the interpretations are sound. The use of SMA model that mimics the human pathology highlights clinical relevance of the outcome. Careful evaluations of motor function, respiratory functions and liver toxicity parameters strengthen their claim that this approach would overcome limitations associated with first-generation vector.

When the authors used three different vector dose, the results indicate that a dose of 1.1×10^{14} demonstrated the highest therapeutic efficacy. This vector dose is almost similar to the dose associated with safety concerns for severe liver injury and acute liver failure. While use of endogenous promoter may overcome unregulated expression of the SMN1 gene, vector capsid-associated toxicity, especially immune response maybe a concern. The authors are encouraged to provide results of anti-AAV9 antibodies and T cells in early and late stages. Overall, this is an interesting contribution to the field.

Referee #2 (Comments on Novelty/Model System for Author):

Although observations from AAV-gene therapy studies from mice are not always fully reliably translatable to humans (requiring e.g. the use of non-human primates on the road to full clinical development) the extent to which AAV9-SMN1 gene therapy in SMA mouse models recapitulates the situation in humans has been extensively described before (not in direct comparisons) and overall observations from mouse models have been highly translatable to patients.

Referee #2 (Remarks for Author):

The current manuscript describes the use of the endogenous SMN1 promoter to drive AAV9-induced SMN1 expression in a mouse model of SMA. The use of general, high-expressing (and unregulated) promoters in gene therapy is a reason for concern about the long-term safety and efficacy of these drugs. In SMA specifically, at least 2000 patients have now been treated with the AAV9-SMN1 gene replacement therapy Zolgensma. Although generally safe and effective, the number of reported SAEs (mostly related to liver toxicity and rare cases of thrombotic microangiopathy) is significant, and treatment outcomes / efficacy varies significantly between patients. There is some concern in the SMA field about the long-term effects of sustained, high-level expression of SMN in especially post-mitotic cells that Zolgensma causes and acute effects on liver function after treatment. In the current manuscript, Xie et al provide a possible solution to this issue by replacing the strong, aspecific cytomegalovirus enhancer/chicken β -actin promoter by a sequence based on the human endogenous SMN1 promoter. The authors's results support their hypothesis that this would be a great solution to toxicity and enhance efficacy. Really it's such an obvious and elegant approach that one wonders why no one has done this before.

I believe this manuscript addresses an important and timely issue in the SMA field and could also serve as a blueprint for gene therapy development for other diseases.

I have a few minor comments:

- I think it would be useful to discuss in more detail the promoter sequence used to design the 2nd generation gene therapy with. The SMN genetic locus is extremely complex and to my knowledge the promoter has been studied in relatively little detail (the authors refer to the only paper that I know that discusses this, which is from 1999). Perhaps the authors could spend some time comparing these sequences also to more recent versions of the human genome such as the recent T2T assembly, and add extra information to their manuscript about their considerations when designing the 2nd generation vector.
- I am always a bit sad when authors still use alpha-tubulin instead of a total protein stain as a loading control (a-tub is also effected by changes in SMN levels -and changed expression of many other proteins too) although the differences observed by the authors are very pronounced so their conclusions can still be supported by normalization to a-tub. It would of course be critical to be able to also see uncropped versions of the western blots in the final version of the manuscript (or supplemental, but I couldn't find them amongst the reviewer files now) as especially those presented in figure 6 seem to have unusually high background levels.
- The microscopy in figure 6 is a bit difficult to interpret due to the small size, it would be useful to add some high-magnification insets to the figure.
- There quite a significant body of literature describing SMN expression levels in various organs and in various mouse models of SMA. It would support the current manuscript to draw some comparisons between the observed expression after 2nd generation

vector treatment and previously observed SMN levels and variation between tissues. Based on these previous studies, it would also be interesting to compare the currently included western blots in figure 6 to SMN levels in especially kidney and spleen at day P30 and P90 (if the tissue is still available).

- I think it would be acceptable to add some more speculative text to the discussion that highlights the challenges of performing trials or other clinical studies with new or improved versions of current SMA therapeutics in the current therapeutic landscape in which many patients are receiving treatment quickly after birth so the number of patient available for trials to compare improved versions of SMA drugs to currently available benchmark drugs might be quite challenging.

Referee #3 (Comments on Novelty/Model System for Author):

The authors use an appropriate model, but do not indicate the sex of the mice, or if any differences in response were noted between sexes. The authors also did not indicate if the data was collected and analyzed in a blinded manner

Referee #3 (Remarks for Author):

Gene therapy is generally held to be the most promising therapeutic option for treatment of spinal muscular atrophy, however there are several concerns associated with the only approved gene therapy, onasemnogene abeparvovec, which delivers an exogenous SMN1 gene. Xie et al. have improved upon onasemnogene abeparvovec by engineering a similar vector that delivers a codon-optimized SMN1 driven by the SMN1 promoter. This second-generation vector restored SMN expression close to physiological levels in a mouse model of SMA, increasing survival, motor, cardiac and respiratory function. Importantly, no increase in transaminase levels were seen, which is a common adverse effect of onasemnogene abeparvovec.

Overall, the conclusions in this manuscript are justified based on the data presented. Improving upon the currently available gene therapy for treatment of SMA, especially with respect to the safety profile, is of the utmost importance. A few points within the manuscript need to be clarified (as outlined below).

Specific points:

The authors did not indicate whether they treated male or female mice, and whether there were any observed sex differences. The authors also did not indicate if any of the data was collected in a blinded manner.

Page 3, paragraph 2 "functional SMN2 transcript" - it is probably more accurate to state "SMN2-derived transcripts that retain exon 7", as even the delta7 transcripts are still functional and give rise to a protein.

Page 4, paragraph 2: With respect to the long-term study in SMA mice suggesting rAAV9-mediated overexpression of hSMN leads to proprioceptive neurons, it should be mentioned that the CMVen/CB promoter was not used in that study, but another ubiquitous promoter.

Page 5, paragraph 2: "Neuro2a-cells" should say "Neuro2a cells" (remove hyphen). Throughout the manuscript, Neuro2A (page 6), Neuro2a (pages 14,15) and Neuro-2a (page 18) are used. Cell line name should be consistent throughout.

Page 6, paragraph 1: How much greater was the expression of SMN in liver tissue of vector 1-treated and benchmark vector-treated mice compared to untreated control? In general, expression levels should be expressed relative to untreated SMA mice and heterozygous mice, not just the benchmark vector.

Page 6, paragraph 3: Please provide further detail about what is meant by "derivative of the endogenous promoter of the hSMN1 gene".

Page 7, paragraph 2: Indicating in the text that all healthy carriers were able to right by day 3 and that untreated mice began to right at day 7 may emphasize the significance of 2nd-generation vector-treated mice righting as soon as day 3 and benchmark vector-treated mice starting to right at day 7.

Page 8, paragraph 1: "2nd-generation vector-treated animals performed as well as healthy carriers on grid test" should specify that this is at day 11 specifically, as the healthy carriers performed better on the grid test at days 9 and 10.

Page 9, paragraph 1: "with a lower respiratory rate" perhaps is not referencing the data in supplementary figure 3, in which the frequency (breaths/minute) between benchmark and 2nd-gen vector appears to be very similar, and no statistics are reported indicating that there is a difference between these two treatment groups.

Page 9, paragraph 4: Including in the text that SMN protein expression in the liver of benchmark vector-treated mice on day 3

was 64-fold above that of healthy carrier may emphasize the degree of supraphysiological SMN expression.

Page 9, paragraph 4: "treatedmice" should be "treated mice"

Page 9, paragraph 4: As SMN expression in the liver of vector-treated mice appears to decrease as time continues, it would be informative to include data about SMN expression in the liver at 60 and 90 days post-treatment, as SMN expression may continue to decrease below physiological levels of healthy carriers.

Page 11, paragraph 1: Why were these organs chosen? Why was liver tissue, or skeletal muscle more relevant to respiration such as the diaphragm, not included?

Page 11, paragraph 1: This paragraph could be improved by comparing SMN expression fold-change between tissues from healthy carriers and 2nd-generation vector-treated mice.

Page 28 figure 1A, h - the codon optimized SMN gene is referred to as "co-hSMN1" in the text of the manuscript, but "opt-hSMN1" in the vector maps - a consistent designation should be utilized through the manuscript.

Page 28 figure 1g, j - "heathy" should be "healthy"

Page 28 figure 1k - "diploit" should be "diploid"

Page 31, figure 4e - can the authors please explain why some of the blots in this image are spliced while others are not. There is a concern that the samples from the different treatments were analyzed on different blots, making comparison between the treatment groups impossible.

Thank you for the submission of your manuscript to EMBO Molecular Medicine and please accept my apologies for the delay in getting back to you, which is due to the fact that one referee needed more time to complete his/her review. We have now received feedback from the three reviewers who agreed to evaluate your manuscript. As you will see from their reports pasted below all three referees support publication of your manuscript raising important but minor concerns. Therefore, I am pleased to inform you that we will be able to accept your manuscript pending the following final amendments:

1) Please address all referee concerns. No additional experiments are required.

Please see the point-to-point response to the reviewers

2) Reorder manuscript sections: Abstract, Introduction, The Paper Explained, Results, Discussion, Materials and Methods, Acknowledgements, Disclosure and competing interests statement, For More Information, References, Figure legends, Tables and their legends, Expanded View Figure legends

Re-ordered as the instruction

3) Figures:

- Please remove all figure from the manuscript text and upload individual, high-resolution file for each main and EV figure. Suppl. Figures 2-5 should be made Figure EV 1-4 and the legends should stay in the manuscript. Suppl. Fig. 1 would best be made an appendix figure - uploaded in PDF format, with a ToC, legend removed from manuscript text and added underneath the figure, and the figure renamed "Appendix Figure S1". Please also update all figure callouts in the main text. For more information on figure presentation please check "Author Guidelines". <https://www.embopress.org/page/journal/17574684/authorguide#datapresentationformat>

Updated

- When presenting images of western blots from different membranes please make sure to clearly indicate it, e.g. with white dotted line, to prevent any doubt of splice sites. This needs to be consistent throughout all the western blots. For example, splice sites are not clear in Figure 4E and Figure 6B.

The sliced images are labeled as dot lines.

4) Author checklist: Please submit a complete checklist. <https://www.embopress.org/pb-assets/embo-site/EMBO%20Press%20Author%20Checklist-1642513524327.xlsx>

5) In the main manuscript file, please do the following:

- Please address all comments suggested by our data editors listed below:

o DAS: Please note that the data availability statement is not provided in the manuscript.

Included

o Figure legends:

1. The legend for figures 3a-c is incorrectly labelled as 3a-b in the manuscript. This needs to be rectified.

Corrected

2. Please note that in figures 1i, k; 4a, d, f; 5a-c; there is a mismatch between the annotated p values in the figure legend and the annotated p values in the figure file that should be corrected.

Corrected

3. Please note that information related to n is missing in the legends of figures 1i, k; 3a-c; 4d; 6a-b, supplementary figure 3.

Included

4. Please note that the error bars are not defined in the legends of figures 1i, k.

included

5. Please note that scale bar and its definition are missing for figures 4c; 6c-d, supplementary figures 4; 5a-b.

Included

6. Please note that in figure 1f; the scale bar unit should be corrected from μM to μm .

Corrected

- Add up to 5 keywords.

SMA, AAV, gene therapy, SMN1 promoter

- Remove data not shown on p.10 and 12.

Removed

- In M&M, provide the antibody dilutions that were used for each antibody.

Included

- In M&M, statistical paragraph should reflect all information that you have filled in the Authors Checklist, especially regarding randomization, blinding, replication etc.

Included

- Add "Disclosure Statement & Competing Interests". We updated our journal's competing interests policy in January 2022 and request authors to consider both actual and perceived competing interests. Please review the policy <https://www.embopress.org/competing-interests> and update your competing interests if necessary.

Added

- Author contributions: Please remove it from the manuscript and specify author contributions in our submission system. CRediT has replaced the traditional author contributions section because it offers a systematic machine-readable author contributions format that allows for more effective research assessment. You are encouraged to use the free text boxes beneath each contributing author's name to add specific details on the author's contribution. More information is available in our guide to authors:

<https://www.embopress.org/page/journal/17574684/authorguide#authorshipguidelines>

Will do during submission

- Please add data availability statement. If no data are deposited in public repositories, add the sentence: This study includes no data deposited in external repositories.

Added.

- Correct the reference citation in the text and reference list. In the text a reference should be cited by author and year of publication. Include a space between a word and the opening parenthesis of the reference that follows. In the reference list, citations should be listed in alphabetical order. Where there are more than 10 authors on a paper, 10 will be listed, followed by "et al.". Please check "Author Guidelines" for more information.

<https://www.embopress.org/page/journal/17574684/authorguide#referencesformat>

Updated

6) Movies: Rename the files to Movie EV1, etc. Remove their legends from the main manuscript file and zipp each legend as a readme.txt file with its corresponding movie.

7) The Paper Explained: Please provide "The Paper Explained" and add it to the main manuscript text. Please check "Author Guidelines" for more information.

<https://www.embopress.org/page/journal/17574684/authorguide#researcharticleguide>

Included

8) Synopsis: Every published paper now includes a 'Synopsis' to further enhance discoverability. Synopses are displayed on the journal webpage and are freely accessible to all readers. They include separate synopsis image and synopsis text.

- Synopsis image: Please provide a striking image or visual abstract as a high-resolution jpeg file 550 px-wide x (250-400)-px high to illustrate your article.

Included

- Synopsis text: Please provide a short standfirst (maximum of 300 characters, including space) as well as 2-5 one sentence bullet points that summarise the paper as a .doc file. Please write the bullet points to summarise the key NEW findings. They should be designed to be complementary to the abstract - i.e. not repeat the same text. We encourage inclusion of key acronyms and quantitative information (maximum of 30 words / bullet point). Please use the passive voice.

Please see below

The Paper Explained

Problem:

Spinal muscular atrophy (SMA) is the most common genetic cause of death in childhood. The disease is caused by loss of function mutations in the survival motor neuron 1 gene (SMN1) encoding SMN. Onasemnogene abeparvovec (Zolgensma®) is an approved adeno-associated virus 9 (AAV9) vector ubiquitously expressing a human SMN1 cDNA transgene for infants with SMA, however, lack of efficacy and adverse events have been observed in patients following treatment.

Results:

We developed a second-generation AAV9 gene therapy expressing a codon-optimized hSMN1 transgene driven by a promoter derived from the native hSMN1 gene. This vector restored SMN expression close to physiological levels in the central nervous system and major systemic organs of a severe SMA mouse model. In a head-to-head comparison between the second-generation vector and a benchmark vector, identical in design to onasemnogene abeparvovec, the second-generation vector showed better safety and improved efficacy.

Impact:

The second-generation vector offers substantial therapeutic benefits over onasemnogene abeparvovec in SMA mice and potentially in SMA patients. The successful use of a regulatory cassette derived from the endogenous SMN1 promoter suggests that endogenous gene-specific promoters for gene replacement or gene silencing holds promise for safer and more efficacious AAV gene therapy.

9) For more information: This space should be used to list relevant web links for further consultation by our readers. Could you identify some relevant ones and provide such information as well? Some examples are patient associations, relevant databases, OMIM/proteins/genes links, author's websites, etc...

10) As part of the EMBO Publications transparent editorial process initiative (see our Editorial at <http://embomolmed.embopress.org/content/2/9/329>), EMBO Molecular Medicine will publish online a Review Process File (RPF) to accompany accepted manuscripts. This file will be

published in conjunction with your paper and will include the anonymous referee reports, your point-by-point response and all pertinent correspondence relating to the manuscript. Let us know whether you agree with the publication of the RPF and as here, if you want to remove or not any figures from it prior to publication. Please note that the Authors checklist will be published at the end of the RPF.

Noted

11) Please provide a point-by-point letter INCLUDING my comments as well as the reviewer's reports and your detailed responses (as Word file).

Please see below

I look forward to reading a new revised version of your manuscript as soon as possible.

Yours sincerely,

Zeljko Durdevic

***** Reviewer's comments *****

Referee #1 (Comments on Novelty/Model System for Author):

An appropriate mouse model of SMA was used in the study.

Referee #1 (Remarks for Author):

This manuscript by Xie et al demonstrates the potential of codon-optimized SMN1 gene expression regulated by an endogenous promoter via an adeno-associated virus vector for improved gene therapy application for spinal muscular atrophy (SMA). SMA is a deadly disease with no long-term therapy until recently when recombinant adeno-associated virus vector (AAV)-based therapeutics has demonstrated efficacy. Nonetheless, vector associated immune response and unregulated transgene expression are encountered. Data provided in this manuscript indicate the potential of codon-optimized gene expression using its own promoter can overcome the limitations. The experiments have been conducted well and the interpretations are sound. The use of SMA model that mimics the human pathology highlights clinical relevance of the outcome. Careful evaluations of motor function, respiratory functions and liver toxicity parameters strengthen their claim that this approach would overcome limitations associated with first-generation vector.

When the authors used three different vector dose, the results indicate that a dose of 1.1×10^{14} demonstrated the highest therapeutic efficacy. This vector dose is almost similar to the dose associated with safety concerns for severe liver injury and acute liver failure. While use of endogenous promoter may overcome unregulated expression of the SMN1 gene, vector capsid-associated toxicity, especially immune response maybe a concern. The authors are encouraged to provide results of anti-AAV9 antibodies and T cells in early and late stages. Overall, this is an

interesting contribution to the field.

We appreciate the positive remarks from the reviewer. Unfortunately, we do not have the material available for T cells testing. We tested the anti-AAV9 antibodies from both vector-treated mice in early stage and found no difference.

Referee #2 (Comments on Novelty/Model System for Author):

Although observations from AAV-gene therapy studies from mice are not always fully reliably translatable to humans (requiring e.g. the use of non-human primates on the road to full clinical development) the extent to which AAV9-SMN1 gene therapy in SMA mouse models recapitulates the situation in humans has been extensively described before (not in direct comparisons) and overall observations from mouse models have been highly translatable to patients.

Referee #2 (Remarks for Author):

The current manuscript describes the use of the endogenous SMN1 promotor to drive AAV9-induced SMN1 expression in a mouse model of SMA. The use of general, high-expressing (and unregulated) promoters in gene therapy is a reason for concern about the long-term safety and efficacy of these drugs. In SMA specifically, at least 2000 patients have now been treated with the AAV9-SMN1 gene replacement therapy Zolgensma. Although generally safe and effective, the number of reported SAEs (mostly related to liver toxicity and rare cases of thrombotic microangiopathy) is significant, and treatment outcomes / efficacy varies significantly between patients. There is some concern in the SMA field about the long-term effects of sustained, high-level expression of SMN in especially post-mitotic cells that Zolgensma causes and acute effects on liver function after treatment.

In the current manuscript, Xie et al provide a possible solution to this issue by replacing the strong, aspecific cytomegalovirus enhancer/chicken β -actin promoter by a sequence based on the human endogenous SMN1 promoter. The authors's results support their hypothesis that this would be a great solution to toxicity and enhance efficacy. Really it's such an obvious and elegant approach that one wonders why no one has done this before.

I believe this manuscript addresses an important and timely issue in the SMA field and could also serve as a blueprint for gene therapy development for other diseases.

Many thanks for the positive comments and suggestions. Actually, the same strategy using endogenous promoter of SMN1 gene for the treatment of SMA was discussed by Dr. Brunhilde Wirth (Wirth 2021) when we were testing the 2nd-generation vector. Her paper was cited in the Discussion as ref 65 in the first submission.

I have a few minor comments:

- I think it would be useful to discuss in more detail the promoter sequence used to design the 2nd generation gene therapy with. The SMN genetic locus is extremely complex and to my knowledge the promoter has been studied in relatively little detail (the authors refer to the only paper that I know that discusses this, which is from 1999). Perhaps the authors could spend some time comparing these sequences also to more recent versions of the human genome such as the recent T2T assembly and add extra information to their manuscript about their considerations when designing the 2nd generation vector.

We thank the reviewer for this thoughtful comment. We examined the relatively new T2T build as recommended and noted that there were only ten differences within the -767 to +152 region used in our vector design and the T2T genome build. We have now reported these differences in a new supplementary figure (**Appendix Figure S2B**). Furthermore, we also found that the transcriptional start site (TSS) for SMN1 is only 3 nt shifted 5' of the original TSS reported. Thus, we believe that our design does indeed encompass the bona fide endogenous promoter. Furthermore, we observed the epigenetic marks across this region reported by the ENCODE Consortium and noted that the -767 to +152 region fully encompasses the marks found within 2 kb upstream of the TSS across all cell types currently reported. We do note that the up-to-date epigenetic tracks for the T2T build are mainly epithelial and cancer cell types. In **Appendix Figure S2A**, we have now included tracks for neuroblasts (BE2C), the only relevant cell type for the T2T build. The rationale of promoter selection is also discussed in the Methods.

- I am always a bit sad when authors still use alpha-tubulin instead of a total protein stain as a loading control (a-tub is also effected by changes in SMN levels -and changed expression of many other proteins too) although the differences observed by the authors are very pronounced so their conclusions can still be supported by normalization to a-tub. It would of course be critical to be able to also see uncropped versions of the western blots in the final version of the manuscript (or supplemental, but I couldn't find them amongst the reviewer files now) as especially those presented in figure 6 seem to have unusually high background levels.

We apologize for not using proper loading control. We conducted an additional Western Blot analysis on the heart samples on Day 30 using a-tub, Gapdh and total protein as loading controls. As the review pointed out, the conclusion remains the same (Fig 1). The uncropped images of figure 6 are shown as Fig 2. Please note when we developed the 2nd-generation AAV-SMN1 which is a self-complementary AAV vector, we were also testing a single-stranded AAV genome with endogenous promoter (ssAAV.SMNp-SMN1). The ssAAV.SMNp-SMN1 vector works more efficiently than the benchmark vector but less than the 2nd-generation vector. To simplify the story, we did not include the ssAAV.SMNp-SMN1 data in this manuscript.

WB Data of 3 month, facial vein, 3.33E+14vg/kg original picture **Brain**

Fig 2

WB Data of 3 month, facial vein, 3.33E+14vg/kg original picture **Heart and Quadriceps**

Fig 2

WB Data of 3 month, facial vein, 3.33E+14vg/kg original picture **Liver**

Fig 2

WB Data of 3 month, facial vein, 3.33E+14vg/kg original picture **Spinal Cord**

Fig 2

- The microscopy in figure 6 is a bit difficult to interpret due to the small size, it would be useful to add some high-magnification insets to the figure.

We have included some high-magnification images in figure 6.

- There quite a significant body of literature describing SMN expression levels in various organs and in various mouse models of SMA. It would support the current manuscript to draw some comparisons between the observed expression after 2nd generation vector treatment and previously observed SMN levels and variation between tissues. Based on these previous

studies, it would also be interesting to compare the currently included western blots in figure 6 to SMN levels in especially kidney and spleen at day P30 and P90 (if the tissue is still available).

Unfortunately, AAV9 by IV does not transduce kidney and spleen efficiently and we did not harvest these tissues as well.

- I think it would be acceptable to add some more speculative text to the discussion that highlights the challenges of performing trials or other clinical studies with new or improved versions of current SMA therapeutics in the current therapeutic landscape in which many patients are receiving treatment quickly after birth so the number of patient available for trials to compare improved versions of SMA drugs to currently available benchmark drugs might be quite challenging.

Thanks for the suggestion. We have included the potential challenges in Discussion. "Notably, performing clinical trials with the 2nd-generation vector can be challenging in Type-1 SMA patients where the birth screening and onasemnogene abeparvovec are available."

Referee #3 (Comments on Novelty/Model System for Author):

The authors use an appropriate model, but do not indicate the sex of the mice, or if any differences in response were noted between sexes. The authors also did not indicate if the data was collected and analyzed in a blinded manner

Both male and females are used in the study and no differences were observed on therapeutic outcomes between genders. The animal study was not done in a blinded manner. We included this information in the revised manuscript.

Referee #3 (Remarks for Author):

Gene therapy is generally held to be the most promising therapeutic option for treatment of spinal muscular atrophy, however there are several concerns associated with the only approved gene therapy, onasemnogene abeparvovec, which delivers an exogenous SMN1 gene. Xie et al. have improved upon onasemnogene abeparvovec by engineering a similar vector that delivers a codon-optimized SMN1 driven by the SMN1 promoter. This second-generation vector restored SMN expression close to physiological levels in a mouse model of SMA, increasing survival, motor, cardiac and respiratory function. Importantly, no increase in transaminase levels were seen, which is a common adverse effect of onasemnogene abeparvovec.

Overall, the conclusions in this manuscript are justified based on the data presented. Improving upon the currently available gene therapy for treatment of SMA, especially with respect to the safety profile, is of the utmost importance. A few points within the manuscript need to be clarified (as outlined below).

We sincerely appreciate the positive comments and the advice to improve the manuscript.

Specific points:

The authors did not indicate whether they treated male or female mice, and whether there were any observed sex differences. The authors also did not indicate if any of the data was collected in a blinded manner.

We have included the gender information in the text and clarified that the study was not done in

a blinded way in Method.

Page 3, paragraph 2 "functional SMN2 transcript" - it is probably more accurate to state "SMN2-derived transcripts that retain exon 7", as even the delta7 transcripts are still functional and give rise to a protein.

Many thanks for the suggestion.

Page 4, paragraph 2: With respect to the long-term study in SMA mice suggesting rAAV9-mediated overexpression of hSMN leads to proprioceptive neurons, it should be mentioned that the CMVen/CB promoter was not used in that study, but another ubiquitous promoter.

We thank the reviewer for making the description accurate.

Page 5, paragraph 2: "Neuro2a-cells" should say "Neuro2a cells" (remove hyphen). Throughout the manuscript, Neuro2A (page 6), Neuro2a (pages 14,15) and Neuro-2a (page 18) are used. Cell line name should be consistent throughout.

We thank the reviewer to point this out.

Page 6, paragraph 1: How much greater was the expression of SMN in liver tissue of vector 1-treated and benchmark vector-treated mice compared to untreated control? In general, expression levels should be expressed relative to untreated SMA mice and heterozygous mice, not just the benchmark vector. (related to Xiupeng)

The increase of SMN1 expression in liver from vector-1 and benchmark vector is greater 49- and 14-fold compared to the untreated control. We have revised the text and Fig 1d.

Page 6, paragraph 3: Please provide further detail about what is meant by "derivative of the endogenous promoter of the hSMN1 gene".

Please see the revised text.

Page 7, paragraph 2: Indicating in the text that all healthy carriers were able to right by day 3 and that untreated mice began to right at day 7 may emphasize the significance of 2nd-generation vector-treated mice righting as soon as day 3 and benchmark vector-treated mice starting to right at day 7.

We appreciate the suggestion and have incorporated it into the main text.

Page 8, paragraph 1: "2nd-generation vector-treated animals performed as well as healthy carriers on grid test" should specify that this is at day 11 specifically, as the healthy carriers performed better on the grid test at days 9 and 10.

Correct it in the text.

Page 9, paragraph 1: "with a lower respiratory rate" perhaps is not referencing the data in supplementary figure 3, in which the frequency (breaths/minute) between benchmark and 2nd-gen vector appears to be very similar, and no statistics are reported indicating that there is a difference between these two treatment groups.

Many thanks for pointing out this error. We have revised the text.

Page 9, paragraph 4: Including in the text that SMN protein expression in the liver of benchmark vector-treated mice on day 3 was 64-fold above that of healthy carrier may emphasize the degree of supraphysiological SMN expression.

We thank the reviewer for the advice. We have included it in the text.

Page 9, paragraph 4: "treatedmice" should be "treated mice"
Corrected it.

Page 9, paragraph 4: As SMN expression in the liver of vector-treated mice appears to decrease as time continues, it would be informative to include data about SMN expression in the liver at 60 and 90 days post-treatment, as SMN expression may continue to decrease below physiological levels of healthy carriers.

We do not have liver samples for day 60, but we have included the day 90 data.

Page 11, paragraph 1: Why were these organs chosen? Why was liver tissue, or skeletal muscle more relevant to respiration such as the diaphragm, not included?

These tissues were selected because of the tropism of AAV9. We regret very much we did not collect diaphragm for analysis

Page 11, paragraph 1: This paragraph could be improved by comparing SMN expression fold-change between tissues from healthy carriers and 2nd-generation vector-treated mice.
Many thanks for the suggestion. The context has been revised.

Page 28 figure 1A, h - the codon optimized SMN gene is referred to as "co-hSMN1" in the text of the manuscript, but "opt-hSMN1" in the vector maps - a consistent designation should be utilized through the manuscript.

Change the opt-hSMN1 in figure 1 to co-hSMN1.

Page 28 figure 1g, j - "heathy" should be "healthy"
Thank you for the correction.

Page 28 figure 1k - "diploit" should be "diploid"

Corrected it.

Page 31, figure 4e - can the authors please explain why some of the blots in this image are spliced while others are not. There is a concern that the samples from the different treatments were analyzed on different blots, making comparison between the treatment groups impossible.

At the beginning of study, we tested two vectors with endogenous promoters, one is a single-stranded AAV vector with 2.1 kb promoter and the other is a self-complementary AAV vector with 0.9 kb promoter (2nd-generation vector). We did WB analysis for the mice treated for 3-, 8- and 12 days. The ssAAV group was loaded in the middle of the gel. After we completed the 90-day study, we found that 2nd-generation vector outperforms the ssAAV, then we did a 30-day study to analyze the heart and respiration functions on the mice received benchmark vector or 2-nd generation vector. In the 30-day study, we did not test the ssAAV vectors. This is why the blot of day3, 8 and 12 were spliced, but not day 30. All the samples are loaded in the same gel for comparison.

1st Feb 2024

Dear Dr. Xie,

We are pleased to inform you that your manuscript is accepted for publication and is now being sent to our publisher to be included in the next available issue of EMBO Molecular Medicine.
